# Integrative modeling of tumor genomes and epigenomes for enhanced cancer diagnosis by cell-free DNA

Mingyun Bae[1,16], Gyuhee Kim[1,16], Tae-Rim Lee[2], Jin Mo Ahn[2], Hyunwook Park[1], Sook Ryun Park[3], Ki Byung Song[4], Eunsung Jun[4], Dongryul Oh[5], Jeong-Won Lee[6], Young Sik Park[7], Ki-Won Song[8], Jeong-Sik Byeon[9], Bo Hyun Kim[10], Joo Hyuk Sohn[11,12], Min Hwan Kim[11], Gun Min Kim[11], Eui Kyu Chie[13], Hyun-Cheol Kang[13], Sun-Young Kong[14], Sang Myung Woo[10], Jeong Eon Lee[15], Jai Min Ryu[15], Junnam Lee[2], Dasom Kim[2], Chang-Seok Ki[2], Eun-Hae Cho [2] ✉ & Jung Kyoon Choi [1] ✉

Multi-cancer early detection remains a key challenge in cell-free DNA (cfDNA)-based liquid biopsy. Here, we perform cfDNA whole-genome sequencing to generate two test datasets covering 2125 patient samples of 9 cancer types and 1241 normal control samples, and also a reference dataset for background variant filtering based on 20,529 low-depth healthy samples. An external cfDNA dataset consisting of 208 cancer and 214 normal control samples is used for additional evaluation. Accuracy for cancer detection and tissue-of-origin localization is achieved using our algorithm, which incorporates cancer type-specific profiles of mutation distribution and chromatin organization in tumor tissues as model references. Our integrative model detects early-stage cancers, including those of pancreatic origin, with high sensitivity that is comparable to that of late-stage detection. Model interpretation reveals the contribution of cancer type-specific genomic and epigenomic features. Our methodologies may lay the groundwork for accurate cfDNA-based cancer diagnosis, especially at early stages.

Noninvasive screening by cell-free DNA (cfDNA) holds great promise for multi-cancer early detection[1]. Circulating tumor DNA (ctDNA) reflects tumor-specific genetic and epigenetic alterations[2–7]. Also, ctDNA fragments are physically shorter than normal cfDNA fragments[8]. Hence, multiple approaches have been applied to detect cancer using these specific characteristics of ctDNA. In addition to targeted approaches relying on deep sequencing of recurrent mutations[2], genome-wide methods have also been developed on the basis of DNA methylation patterns[6,7] and genomic fragmentation patterns coupled with copy number variations (CNVs)[9] or with chromatin signatures[10].

Whole-genome sequencing (WGS) has been found to be more sensitive than targeted deep sequencing in detecting low-burden

diseases. Zviran et al.[11] suggested that ultrasensitive monitoring of minimal residual diseases is possible by taking full advantage of the cumulative signal of a large number of mutations. However, this approach only tracks the initial mutation profile of the patient's tumor tissue, instead of identifying mutations de novo. To date, cfDNA WGS has not been attempted for de novo cancer detection, mainly because of inaccurate variant calling and filtering from cfDNA.

Unlike targeted deep sequencing of driver mutations, de novo mutation calling from cfDNA can be of low confidence especially without matched normal control. In this work, to compensate for the lack of matched control, we leverage reference cfDNA sequencing data from a large healthy cohort in the process of filtering out germline

mutations and various types of artifacts. This enables us to capitalize on information embedded in a large amount of passenger mutations. Jiao et al.[12] and Nguyen et al.[13] demonstrated that the distribution of passenger mutations, calculated as regional mutation density, serve as a powerful feature in predicting the origin of cancer. In a recent proof-of-concept study, Wan et al.[14] utilized mutation signatures derived from cfDNA variants for the purpose of cancer detection. As pointed out in this pilot study, the potential of using cfDNA mutation signatures for de novo cancer detection has to be tested in a much larger cohort of samples. Based on these studies, we hypothesize that local mutation density (LMD) and mutation signatures can make the most of cfDNA WGS data. Specifically, we attempt to use cfDNA WGS for multi-cancer detection with the 'genome model' that integrates large-scale reference cfDNA data from our healthy cohort and tumor tissue mutation data from the PCAWG project[15]. In doing so, we anticipate that tumor-derived mutations can be captured by the genome model when coupled with systematic cfDNA variant filtering and large-scale reference database.

Genomic locality is expected for not only variant density but also for ctDNA burden. cfDNA fragments from nucleosome-depleted regions (NDRs) are more frequently degraded in the blood than nucleosome-protected DNA, resulting in nucleosome footprints that reflect tissue type-specific chromatin architecture[16]. Such signatures from promoter NDRs were used to infer gene expression programs of the tissues of origin[17] or estimate ctDNA burden[18]. However, this epigenomic feature has not been explored across the whole genome beyond promoter regions or used for multi-cancer detection by cfDNA. In this work, we attempt to use cfDNA WGS for multi-cancer detection with the 'epigenome model' that integrates pan-cancer whole-genome chromatin profiles based on the Assay for Transposase-Accessible Chromatin using sequencing (ATAC-seq)[19]. To this end, we construct tissue-specific NDR profiles based on ATAC-seq data for TCGA samples of various tumor types.

Previous cfDNA studies have focused on the distribution of read fragments falling across NDRs. In this work, we hypothesize that the sensitivity of cancer detection can be improved by considering fragment length and read distribution simultaneously. In order to incorporate fragment length together with read distribution as a prediction feature, we employ the V-plot that is widely used for the analysis of nucleosome positioning based on ATAC-seq data[20]. In our epigenome model, the V-plot is used to visualize the density of fragment length as a function of the location of fragment centers. Convolutional neural networks (CNNs) are used to process the V-plot profiles as three-dimensional images.

Here, we test the genome model and epigenome model by using cfDNA WGS data for 2125 patient samples of 9 cancer types and 1,241 normal control samples, totaling 3366 samples (Supplementary Data 1). To enable the validation of the robustness of the models, we perform sequencing on two different platforms at different depths in two separate batches (Supplementary Data 2) and also employ the data for 422 publicly available samples used for 'DNA evaluation of fragments for early interception' (DELFI)[9]. The DELFI algorithm along with other existing methods are used to evaluate our models. We show how our algorithm enables accurate early cancer detection and tissue-of-origin localization by incorporating cancer type-specific profiles of mutation distribution and chromatin organization in tumor tissues as model references.

## Results
### Model development
The genome model employed a thorough variant filtering process (Fig. 1). In particular, we performed low-pass WGS on 20,529 healthy normal cfDNA samples. This dataset served as a normal reference panel for filtering potential biological and technical noise against genuine tumor-derived DNA, including clonal hematopoiesis (CH)

variants, germline variants, and sequencing artifacts. Additional germline and artifact filtering was performed on the basis of the public databases. This process tended to filter out variants called from healthy samples while retaining those from patient samples (Supplementary Fig. 1A).

The LMD values were estimated by using 2754 PCAWG[15] samples (Supplementary Data 3) across the whole genome for each cancer type as a reference for the local variant density (LVD) calculated from cfDNA (Fig. 1). Cancer type-specific high LMD regions and low LMD regions were identified (Supplementary Fig. 1B and Supplementary Data 4). Without variant filtering, the cfDNA LVD did not agree with the LMD distribution in matching tissues (Supplementary Fig. 1C left). In contrast, our filtering process resulted in selective detection of cfDNA variants in high LMD regions of the matching cancer type (Supplementary Fig. 1C right). Based on the filtered variants, 2,726 LVD features and 150 variant type features were computed. Deep neural networks were employed for predictive modeling of these genomic features (Fig. 1 left).

To construct the NDR profiles for the epigenome model, we tested the two peak callers, namely HMMRATAC[21] and MACS2[22], by using ATAC-seq data from the GM12878 and K562 cell lines for which a range of histone modification data was also available. As a result, we found that HMMRATAC was capable of calling more distinct NDR peaks than MACS2 was (Supplementary Fig. 2A). We also examined whether the identified NDRs exhibit the expected V-plot patterns[20]. As a result, NDR identification based on HMMRATAC, but not MACS2, produced the expected patterns of sequencing fragments according to the distance from the peak midpoint (Supplementary Fig. 2B). In addition, the bimodal distribution of various types of histone modification supported nucleosome depletion at the NDRs identified using HMMRATAC in contrast to those identified by MACS2 (Supplementary Fig. 2C). Finally, cell type-specific NDR patterns were confirmed between the cell lines when HMMRATAC was used but not when MACS2 was used (Supplementary Fig. 2D).

The epigenome model was developed on the basis of the chromatin accessibility landscape of 431 samples (Fig. 1 and Supplementary Data 5). The ATAC-seq data of 410 TCGA samples from 23 cancer types[19] were employed to profile cancer type-specific cfDNA depletion patterns. To account for confounding effects of cfDNA from peripheral blood mononuclear cells (PBMCs), ATAC-seq profiles of major PBMC types were utilized. The peak calling and processing pipeline was applied to identify tissue-specific NDRs (Supplementary Fig. 2E, F and Supplementary Data 6–7). The cfDNA read data were transformed into three-dimensional V-plot[20] images to visualize the fragment size as a function of the coordinates relative to the NDRs. CNNs were employed for predictive modeling of the image data (Fig. 1 right).

### Model evaluation
To test our algorithms, we generated cfDNA WGS data for a total of 3366 samples on two different sequencing platforms (Supplementary Data 2). The MGI and Illumina platform data were generated at an average of 5× and 2.5× depth, respectively, in two separate batches. Samples from the larger batch served as the training cohort, leaving the remaining batch for validation. In addition, the cfDNA WGS data for 422 publicly available samples used for DELFI development[9] were adopted for further evaluation.

In addition to the individual performance of the genome model and epigenome model, the effect of combining the two models was also assessed. For the combined model, the average of the prediction scores of the two models was obtained. For comparison with the genome, epigenome, and combined models, we implemented predictions based on fragmentation patterns (DELFI)[9], fragment size profiles[8], and copy number variations[3] (Supplementary Data 8–9). The score combining fragmentation with other features, which was provided from the DELFI publication[9], was also compared.

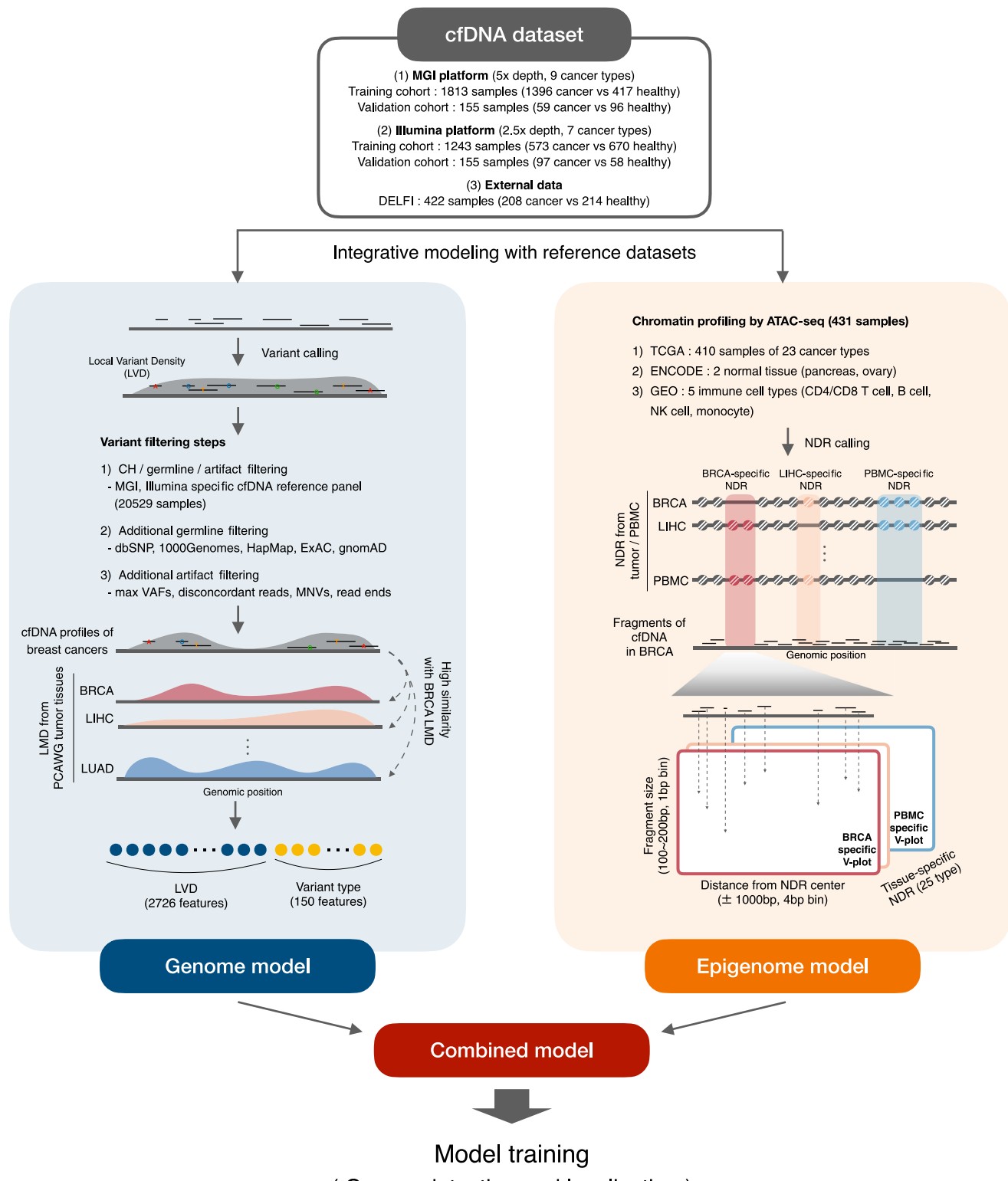

**Fig. 1 | Schematic of the concept and pipeline of our models.** Schematic of the genome model (left) and epigenome model (right) developed based on the integration of reference tissue data for mutation distribution and chromatin architecture, respectively. For illustration, a breast cancer cfDNA sample is used. For the genome model, various variant filtering steps were incorporated. In addition to systematic filtering by our healthy cfDNA-based reference panel, additional processes involving public databases and technical methods were employed. The LVD and variant types were obtained from the filtered set of variants to be used as input to the genome model. To evaluate LVD patterns, cancer type-specific LMD was calculated from 2,754 tissue WGS samples. For the epigenome model, tissue-specific NDRs were identified by processing ATAC-seq data for 431 samples obtained from public databases. cfDNA fragmentation at the identified NDRs was fed into CNNs as three-dimensional V-plot images. Three different cfDNA WGS datasets, totaling 3788 samples, were used to test the genome and epigenome algorithms. Our data were prepared in two separate batches, representing the training and validation cohorts. BRCA Breast cancer, LIHC Liver hepatocellular carcinoma, LUAD Lung adenocarcinoma.

We trained and evaluated our algorithms in each training cohort using stratified five-fold cross-validation (Supplementary Fig. 3A, B). The hyperparameters that were fixed through a hyperparameter optimization process were used for 30 iterations of model training in different random states. From the repetitions of training, the model with the least validation loss value was selected. There was no significant difference in the validation loss among the iterations (Supplementary Figs. 4–5). In addition, the MGI validation cohort and Illumina validation cohort were used to verify the robustness of our models (Supplementary Fig. 3C).

The genome model outperformed all other methods in both MGI cohorts (Fig. 2A, Table 1 and Supplementary Data 8). For the Illumina data, the genome model and epigenome model showed the best performance on the training and validation cohorts, respectively (Fig. 2B). The limited accuracy of the genome model in the Illumina validation cohort may be attributed to relatively low sequencing coverage. In both datasets, the effect of combining the two models was compelling in the training cohort but not well pronounced in the validation cohort, probably due to an insufficient sample size. Nonetheless, the combined model maintained robust performance with a consistent ROC-AUC > 0.9 for all examined cohorts. The DELFI dataset[9] also supported the superiority of our models (Supplementary Fig. 6A, B); the score reported by the authors[9] was comparable to the epigenome model but outperformed by the genome and combined models.

The sensitivity of cancer detection was examined across stages and cancer types (Fig. 2C, D and Supplementary Fig. 7). Whereas other methods performed better for stage III–IV cancers, our methods, especially the genome and combined models, detected stage I–II cancers with a high sensitivity comparable to that of late-stage detection. At 95% specificity, the combined model had a sensitivity of 91.1% on the MGI data and 79.6% on the Illumina data for stage I cancers (Fig. 2C, D). Similar patterns were maintained at 98% and 99% specificity (Supplementary Fig. 7). Cancer types with large amounts of data, such as breast and liver cancers, showed high sensitivity in general. Pancreatic cancer was accurately detected even with a relatively small sample size, especially owing to the epigenome model. The DELFI dataset[9] also proved the superiority of our models in early cancer detection (Supplementary Fig. 6C). At 95% specificity, the combined model had a sensitivity of 98.2% for stage I cancers on this dataset.

The accuracy of localizing the tissue of origin was evaluated for tumor types with $n > 30$ (Fig. 3, Supplementary Fig. 8, and Supplementary Data 9). For both the MGI (Fig. 3A) and Illumina (Fig. 3B) cohorts, the genome and epigenome models outperformed the existing methods, and further improvements were observed when the two models were combined. Again, cancer types with a large number of samples tended to perform better; the localization of pancreatic cancer was relatively more accurate than expected based on the data size (Fig. 3A right, 3B right, and 3C). Similar results were observed when the same evaluation process was repeated for a subset of samples correctly identified from the cancer detection model (Supplementary Fig. 9).

We performed further validation for our models. First, we tested whether the prediction scores of the combined model, genome model, and epigenome model actually correlated the tumor fraction of each sample measured in cfDNA (Supplementary Fig. 10A, B). Second, we performed a type of external validation by testing a model trained on one cohort using data from the other cohorts with different sequencing platforms and experimental procedures. Specifically, the performance of the model trained on the MGI data was measured on the Illumina training cohort, Illumina validation cohort, and DELFI cohort (Supplementary Fig. 10C left). Likewise, the performance of the model trained on the Illumina data was measured on the MGI training cohort, MGI validation cohort, and DELFI cohort (Supplementary Fig. 10C right). In all cases, a consistent ROC-AUC > 0.8 was obtained. Also, the normal and tumor samples of the MGI cohort were clearly segregated by the prediction scores trained with the Illumina cohort, and vice versa (Supplementary Fig. 10D). Third, we checked the robustness of

our model against low read depth by applying downsampling to the WGS data. Although the 1× downsampling resulted in a slightly lower ROC-AUC, comparable levels of performance were obtained between the original 5× and downsampled 3× data with the combined model, genome model, and epigenome model (Supplementary Fig. 10E, F).

## Model interpretation

We wanted to assess how the reference tumor tissue data and reference normal cfDNA data contributed to model performance. First, the effect of variant filtering based on our normal reference panel consisting of 20,529 healthy cfDNA genomes (Fig. 1) was assessed. This filtering process significantly improved the accuracy of cancer detection (Fig. 4A left) and localization (Fig. 4A right) by the genome model, thereby indicating that marking potential non-tumor variants was essential for accurate LVD estimation. Fully using LVD features across the whole genome was most optimal (red plots in Fig. 4B). However, the genome model with selected high or low LMD regions outperformed its counterpart with the same number of random regions with respect to both cancer detection (Fig. 4B left) and localization (Fig. 4B right). These findings indicate that the genome model is capable of capturing the biological aspects of the mutation distribution in the original tumor tissues from the cfDNA variant data. Unlike the genome model, the epigenome model could be developed only with selected regions (Fig. 1). The epigenome model using fragmentation data from tissue-specific NDRs resulted in higher accuracy than its counterpart using fragmentation data from the same number of random regions with respect to both cancer detection (Fig. 4C left) and localization (Fig. 4C right).

We then used feature attribution to measure how much each feature in a prediction model contributes to the predictions for each given instance. For the genome model for cancer detection, we compared the tissue LMD values between the LVD regions with positive attribution and those with negative attribution. As a result, higher LMD values were observed for the regions that were assigned positive attribution in both the MGI cohort (Fig. 5A) and the Illumina cohort (Supplementary Fig. 11A), implying that cfDNA mutations identified in genomic regions with high mutation rates in tumor tissues increase the likelihood of predicting the given sample as cancer.

In the genome model for tumor-site localization, we compared the attribution scores between cancer type-specific LMD high versus low regions (Fig. 5B and Supplementary Fig. 11B, C). Positive attribution assigned to high LMD regions (red plot in Fig. 5B and Supplementary Fig. 11B) indicates that high LVD values of cfDNA samples in these regions of a given cancer type increased the likelihood of the model predicting these samples as the corresponding cancer type. In contrast, when cfDNA samples have high LVD values in low LMD regions of a given cancer type, the genome model tended to not predict these samples as the given cancer type, resulting in negative attribution scores (blue plot in Fig. 5B and Supplementary Fig. 11B). The biased distribution of attribution was not observed for high and low LMD regions of the cancer types that did not match the given prediction label (Fig. 5C and Supplementary Fig. 11D).

Finally, the attribution values assigned to each feature of the genome model were used to cluster the cfDNA samples across the cohorts. The samples were clustered by the label (i.e., tumor or normal) rather than by the cohort in both the cancer detection model (Fig. 5D left) and cancer localization model (Fig. 5D right). These results confirm that the genome model learned biological differences between tumor and normal cfDNA but not technical biases attributed to different sequencing platforms or experimental procedures.

We also examined how the epigenome features contributed to the model in terms of attribution. For the cancer detection epigenome model, we averaged the attribution values along the z axis of the CNN input images (i.e., across the V-plots for tissue types) (Fig. 6A and Supplementary Fig. 12A). When plotted for visual inspection, the

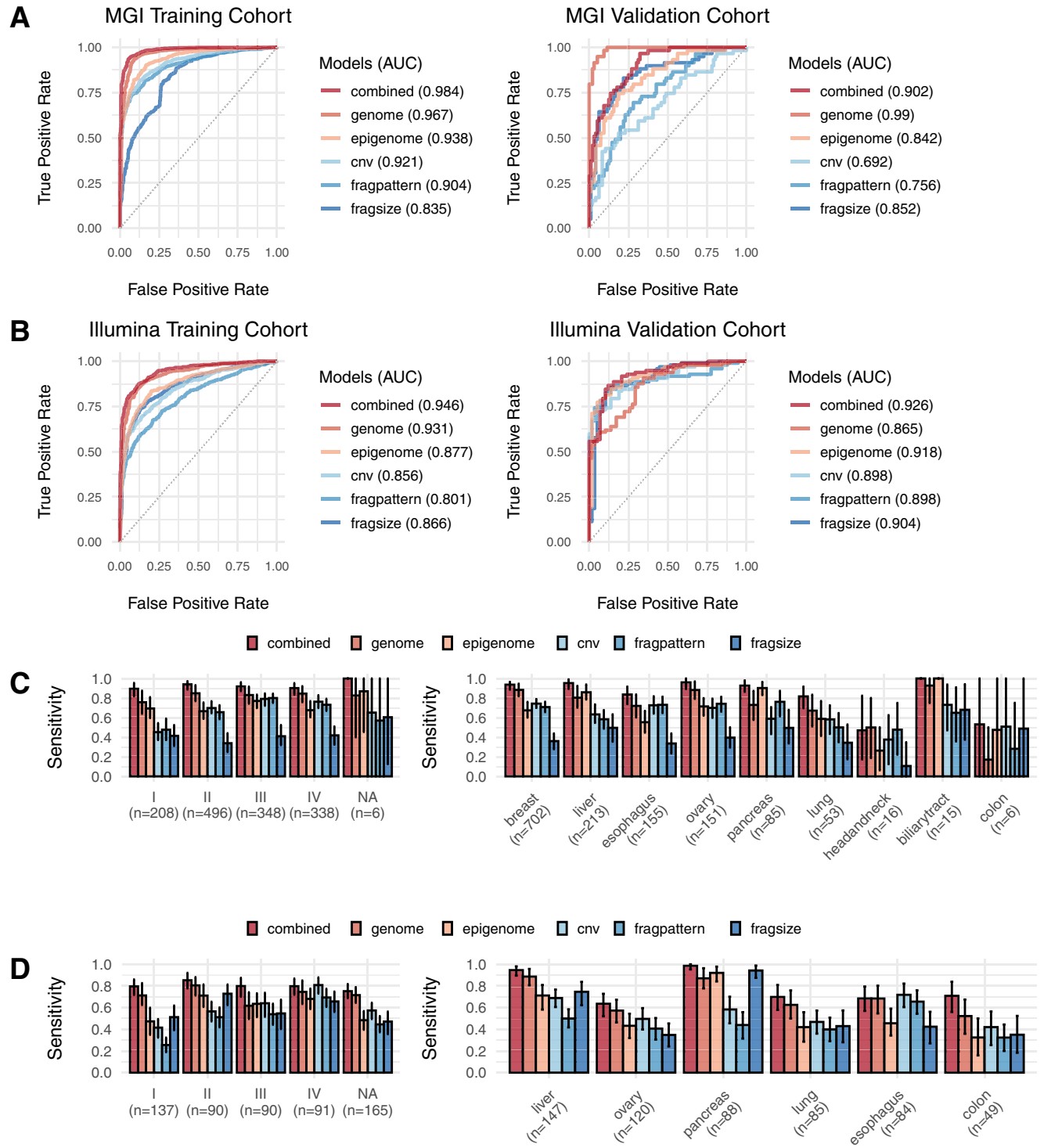

**Fig. 2 | Performance of cancer detection. A** ROC curve for different models on the MGI training cohorts. A total of 1396 cancer and 417 normal control samples were used for training, and 59 cancer and 96 normal control samples were used for validation as the MGI validation cohort. **B** ROC curve for different models on the Illumina training cohort. A total of 573 cancer and 670 normal control samples were used for training, and 97 cancer and 58 normal control samples were used for validation as the Illumina validation cohort. **C, D** Sensitivity values with the 95% confidence interval at 95% specificity broken down by the tumor stage (left) and cancer type (right) for the **C** MGI training cohort and **D** Illumina training cohort. Confidence interval for sensitivity value was calculated from 1000 bootstraping samplings. **A–D** Our genome, epigenome, and combined models were compared with predictions based on fragmentation patterns[9] (fragpattern), fragment size profiles[8] (fragsize), and copy number variations[3] (cnv). NA Stage information not available, CI Confidence interval. Source data are provided as a Source Data file.

average attribution values were biased to shorter fragments specifically in cancer cfDNA samples of the MGI cohort (Fig. 6A, B) and the Illumina cohort (Supplementary Fig. 12A, B).

The same analysis was performed for the epigenome model for cancer localization. A periodic distribution of high and low attribution values was observed according to the distance from the NDR center, more distinctly with the MGI data (Fig. 6C) and rather weakly with the Illumina data (Supplementary Fig. 12C). This periodicity of model attribution seems to reflect regular spacing between positioned nucleosomes surrounding the NDR. While depletion of cfDNA reads at

**Table 1 | Model performance of cancer detection**

| Models | MGI platform | | | | Illumina platform | | | |
|---|---|---|---|---|---|---|---|---|
| | Training Cohort | | Vadliation Cohort | | Training Cohort | | Vadliation Cohort | |
| | auc | 95% CI | auc | 95% CI | auc | 95% CI | auc | 95% CI |
| Combined | 0.984 | 0.978–0.988 | 0.902 | 0.85–0.944 | 0.946 | 0.933–0.958 | 0.926 | 0.882–0.962 |
| Genome | 0.967 | 0.956–0.976 | 0.99 | 0.978–0.998 | 0.931 | 0.916–0.944 | 0.865 | 0.807–0.915 |
| Epigenome | 0.938 | 0.926–0.949 | 0.842 | 0.772–0.899 | 0.877 | 0.857–0.896 | 0.918 | 0.871–0.958 |
| CNV | 0.921 | 0.908–0.933 | 0.692 | 0.608–0.778 | 0.856 | 0.833–0.877 | 0.898 | 0.847–0.942 |
| Fragpattern | 0.904 | 0.889–0.917 | 0.756 | 0.676–0.827 | 0.801 | 0.776–0.826 | 0.898 | 0.839–0.946 |
| Fragsize | 0.835 | 0.813–0.86 | 0.852 | 0.788–0.91 | 0.866 | 0.846–0.887 | 0.904 | 0.848–0.953 |

the NDRs of a particular cancer type should increase the likelihood of assigning the given sample to the corresponding cancer type, enrichment of cfDNA reads at the NDRs of a particular cancer type is supposed to decrease the prediction probability for the given cancer type. As expected, negative attribution scores were assigned to reads mapping to the NDRs of a matching cancer type (red line in Fig. 6D, E and Supplementary Fig. 12D, E).

Finally, the attribution values assigned to each feature of the epigenome model were used to cluster the cfDNA samples across the cohorts. As a result, the samples tend to cluster by the label (i.e., tumor or normal) rather than by the data source for both cancer detection (Fig. 6F left) and tissue-of-origin localization (Fig. 6F right). Taken together, similarly as the genome model, our epigenome model appears to learn biological differences between tumor and normal cfDNA but not technical biases due to different sequencing platforms or experimental procedures.

## Discussion
Early detection of cancers of various types is an important part of cancer medicine since cancer has a better prognosis and survival rate when diagnosed and treated earlier[23,24]. In most cases, especially in pancreatic cancers, diagnostic tests are performed after symptoms arise, and the proper treatment times are often missed. Solutions for this urgent unmet need of multi-cancer early detection and localization are being actively pursued by using cfDNA-based noninvasive cancer screening[1].

Recent studies have demonstrated the power of screening cfDNA at the whole-genome scale[9,11]. Ultrasensitive detection of minimal residual disease was made possible by capitalizing on genome-wide cumulative signals when informed by original tumor profiles[11]. However, de novo detection of cancer, especially at early stages, remains challenging; the sensitivity of the DELFI algorithm for stage I diseases at 95% specificity was reported to be 73%[9] but was even lower when applied to our larger cohorts (below 50%). To exploit the full potential of whole-genomic cfDNA screening, we developed an analytical strategy empowered by a large amount of training and reference data from cfDNA and tissue samples. Integrative modeling that incorporates the knowledge of how tumor genomes and epigenomes leave their footprints in cfDNA shows unprecedented accuracy of cancer detection at a sensitivity of 91.1% and 98.1% (at 95% specificity) for stage I cancers on the MGI and DELFI dataset, respectively.

Our study has several implications. First, we demonstrate how integrating large-scale reference datasets including the genome and epigenome data from tumor tissues and the normal cfDNA data for variant filtering can increases the sensitivity of cancer detection. In particular, this work involves an attempt to analyze cfDNA data based on modeling of cancer type-specific LMD and NDR profiles derived from large-scale public data in tumor tissues. Second, we propose genomic and epigenomic features that are effective for cfDNA-based cancer diagnosis. The genomic features based on LMD and mutation signatures, which proved to be useful in previous tissue-based cancer type

classification, were applied to cfDNA analysis in this work. Our epigenome model employed the V-plot for the integrative analysis of fragment density and length in NDRs. Third, as feature modeling is critical in developing our models, we provide comprehensive characterization of contributing features instead of leaving the models as black boxes. Most of previous cfDNA-based prediction models examined the properties of input features, but did not investigate which features were actually learned during model training and thus contributed most to prediction. In this work, we not only determined which features played an important role in model development and performance, but also investigated the relevance of the features to the tumor biology in terms of genetic and epigenetic characteristics of cancer.

In terms of further improvement in accuracy, we have observed three potential factors that appear to affect the performance of our models. The fact that the genome model generally performs better than the epigenome model may be attributed to the amount of reference tissue data. Increasing the number of cases for ATAC-seq or other epigenome data may provide a more accurate reference for the chromatin architecture. However, the compelling performance of the epigenome model with pancreatic cancer emphasizes the importance of making the most of tumor biology. Another factor is the data size used for model training. In both cancer detection and localization, cancer types with large sample sizes, such as breast and liver cancers, are diagnosed with the highest accuracy. Data size per sample also matters. The overall trend that better performance is achieved on the MGI data compared with the Illumina data implies that higher read depths contribute to prediction accuracy. Hence, when coupled with integrative modeling based on tumor biology, an increasing amount of reference and training data for cfDNA analysis has the potential to realize ultrasensitive early cancer detection and accurate cancer diagnosis.

## Methods
### Characteristics of patient and healthy samples
Plasma samples form healthy individuals and patients with breast, hepatic, lung, ovarian, colorectal, pancreatic, bile duct, esophagus, or head and neck cancer were obtained from GC Genome, Samsung Medical Center, Seoul National University Hospital, Asan Medical Center, National Cancer Center, and Yonsei Cancer Center. All the healthy and patient samples were obtained under Institutional Review Board approved protocols with informed consent from all participants for research use at participating institutions (IRB 2021-161-1184, IRBC 2021-02-070, IRB 2021-0596, IRB 2021-0399, IRB 2021-0399, and IRB 2020-09-002 for patient samples; GCL IRB 2017-1008-03, GCL IRB 2020-1002-04, and GCL IRB 2021-1049-02 for healthy samples). Plasma samples from healthy individuals were obtained if they had no previous history of cancer and negative routine health screening questionnaire. Clinical information for all samples used in model training and evaluation are listed in Supplementary Data 2. Normal reference samples were obtained from noninvasive prenatal screening under Institutional Review Board approved protocol (GCL-2021-1060-01) with anonymization.

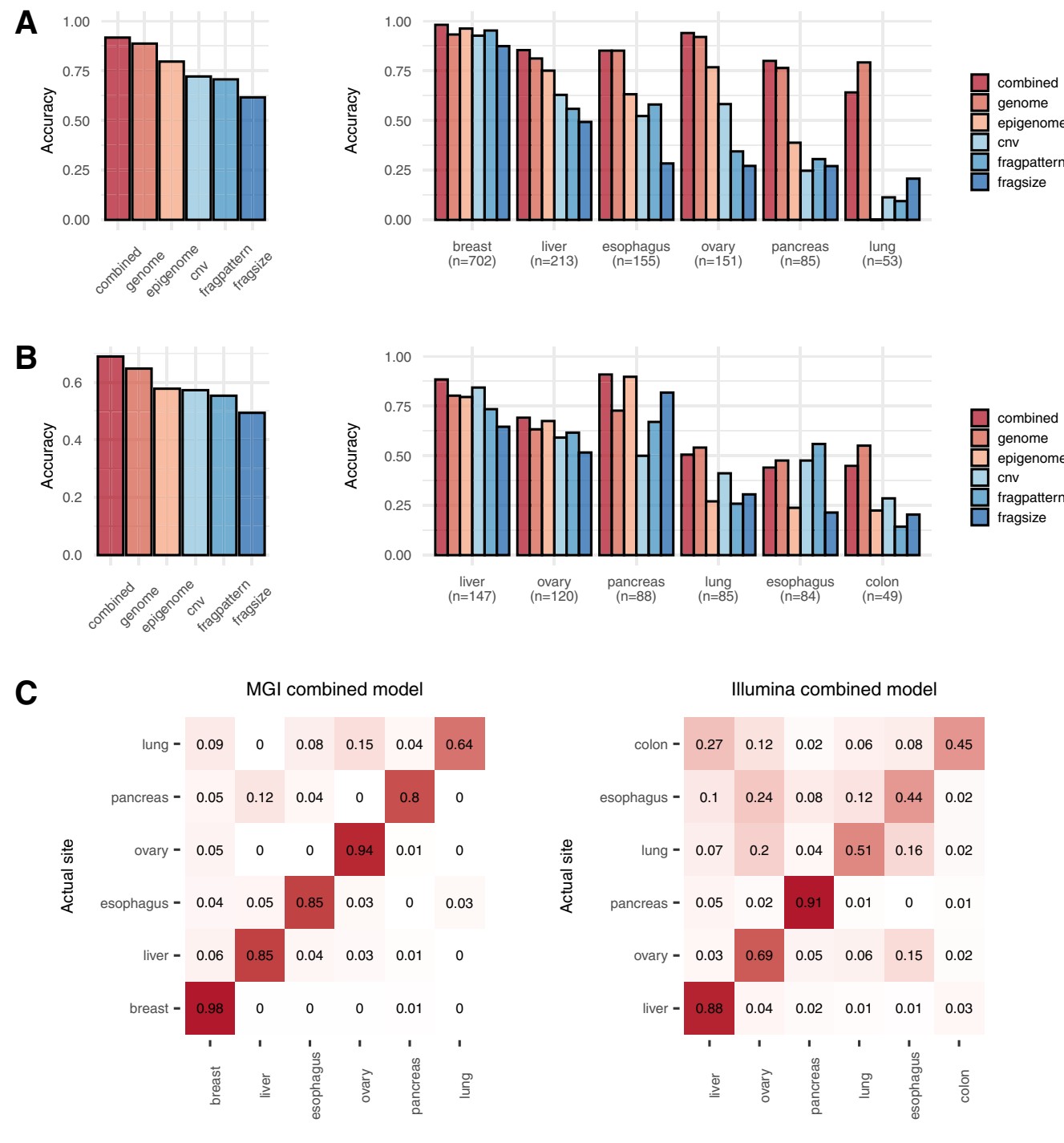

**Fig. 3 | Performance of tissue-of-origin localization. A** Average accuracy (left) and accuracy (right) on each cancer type for different models on the MGI training cohort. **B** Average accuracy (left) and accuracy (right) on each cancer type for different models on the Illumina training cohort. **C** Confusion matrix for localization using the combined model on the MGI cohort (left) and Illumina cohort (right). The y-axis represents the actual site, and the x-axis represents the predicted site.

The numbers in the cells of the matrix represent the proportion of samples of each cancer type localized to respective tumor sites. **A**–**C** Our genome, epigenome, and combined models were compared with predictions based on fragmentation patterns[9] (fragpattern), fragment size profiles[8] (fragsize), and copy number variations[3] (cnv). Source data are provided as a Source Data file.

## cfDNA preparation and sequencing library construction

From each sample, 10 mL of whole blood was collected in Cell-Free DNA BCT tubes (Streck, US). The collected blood samples were centrifuged to separate plasma and cellular components, and plasma was processed immediately or within 24 h.

For sequencing on an MGI platform, cfDNA was extracted from 0.6 mL plasma and eluted in a final volume of 56 μL, using a plasma circulating DNA Kit (Tiangen, China) according to the manufacturer's instructions. Extracted cfDNA was processed for library construction starting with a 2−6 ng input, using the MGIEasy Cell-free DNA Library Prep Kit (MGI, China) according to the manufacturer's instructions. The concentration of the library was quantified using the dsDNA HS Qubit Assay (Invitrogen, US). The size of the library was determined using a D1000 screentape assay with 2200 Tapestation (Agilent

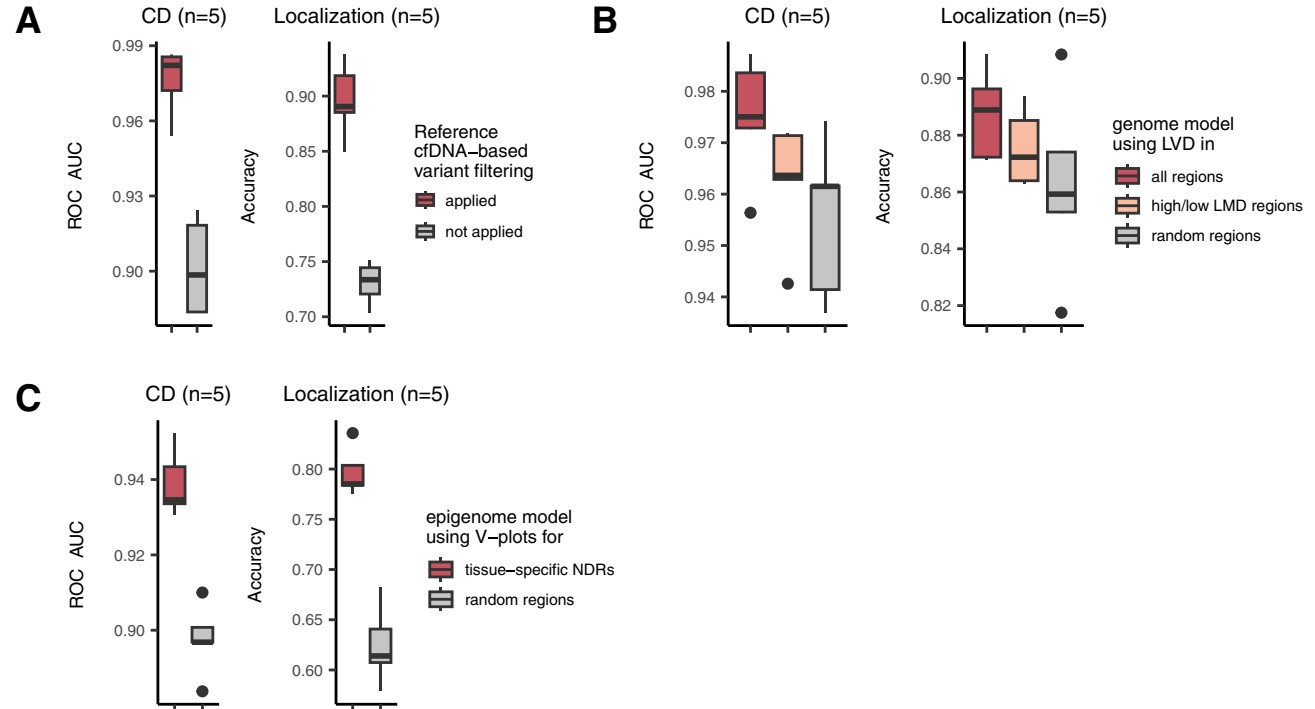

**Fig. 4 | Contributions of the tumor and cfDNA reference data. A** ROC-AUC and accuracy of the genome model depending on whether our reference cfDNA panel was used for cancer detection (left) and localization (right). **B** ROC-AUC and accuracy of the genome model according to which genomic regions were used for LVD feature acquisition in cancer detection (left) and localization (right). The total number of cancer type-specific high or low LMD regions are 264. The same number of random regions were selected for comparison with high or low LMD regions. **C** ROC-AUC and accuracy of the epigenome model depending on which NDRs were used for the construction of V-plot images for cancer detection (left) and localization (right). The same number of random regions were selected for comparison with tissue-specific NDRs. **A**–**C** Five-fold cross-validation was performed to examine the distribution of ROC-AUC and accuracy values. Each box indicates IQR and median, whiskers indicates 1.5× IQR, black dots indicates outlier. CD cancer detection, Localization cancer localization. Source data are provided as a Source Data file.

Technologies, US). The sequencing libraries were pooled, and up to 20 libraries per batch were multiplexed.

For sequencing on an Illumina platform, cfDNA was extracted from 0.4 mL plasma applied with an automated KingFisher system (Thermo-Fisher Scientific, US) and eluted in a final volume of 22 μL, using an Apostle MiniMax High Efficiency cfDNA Isolation Kit (Apostle, US) according to the manufacturer's instructions. Extracted cfDNA was processed for library construction starting with a 0.1–7 ng input, using the Swift 2 S® Sonic DNA Library Kit (IDT, US) according to the manufacturer's instructions. The concentration of the library was quantified using the dsDNA HS Qubit Assay (Invitrogen, US). The size of the library was determined using a D1000 screentape assay with 2200 Tapestation (Agilent Technologies, US). The sequencing libraries were pooled, and up to 240 libraries per batch were multiplexed.

### Processing of the MGI and Illumina cohort and reference cfDNA data

The prepared libraries for the MGI cohort samples were subjected to 100 bp paired end runs on a DNBSEQ-G400 sequencing instrument (MGI, China). The libraries for the Illumina cohort samples were sequenced with 100 bp paired end reads using a NovaSeq 6000 S4 Reagent Kit v1.5 (Illumina, US). The fastq files were aligned to the human hg19 genome using BWA-MEM 0.7.5a. Duplicate marking was performed with the aligned bam files using the Genome Analysis Toolkit (GATK)[25]. Additional processing was conducted differently for the genome and epigenome model as described in the "Genome model input processing and training" and "Epigenome model input processing and training" sections.

The cfDNA libraries of 20,529 normal reference samples were sequenced at an average of 0.3 × 75 bp single ends using NextSeq 500

High Output Kit v2.5 (Illumina, US) or 0.6 × 100 bp paired ends on DNBSEQ-G400RS (MGI, China). The fastq files were trimmed using the Trim Galore tool in Cutadapt[26] with the -clip_R1 10 -clip_R2 10 -length 20 options. The reads were aligned to the human hg19 genome using BWA-MEM 0.7.5a. The aligned bam files were processed using the GATK pipeline of data preprocessing for variant discovery, including duplicate marking, indel realignment, and base quality score recalibration[25]. Because of the low depth and large sample size of the Illumina dataset, duplicates were removed by adding the −REMOVE_DUPLICATES true option when performing duplicate marking, and the low mapping quality reads were removed using the SAMtools -q60 option, instead of performing base quality score recalibration and indel realignment. The Illumina bam files were merged in a batch size of 1000 samples using the merge function in SAMtools, and the merged bam files were subjected to variant calling. Mutect2[25] was used to call variants in the tumor-only mode with the -flr2-max-depth 1000 -initial-tumor-lod 0.1 options for the merged Illumina bam files and -max-mnp-distance 0 option for the individual MGI bam files. After variant calling, we combined all of the variants called from the merged bam files to construct the Illumina normal panel, and implemented the GATK pipeline of creating a somatic panel of normals to construct the MGI normal panel. Finally, the Illumina and MGI panels were combined to create a normal cfDNA reference panel.

### Identification of cancer type-specific LMD high/low regions

We obtained the variant call dataset for tumor tissues from the ICGC portal of the PCAWG project[15] (https://dcc.icgc.org/releases/PCAWG/). All samples listed in the PCAWG exclusion list or in the PCAWG microsatellite instability list were excluded. The remaining 2754 samples were used to identify cancer type-specific LMD high and low

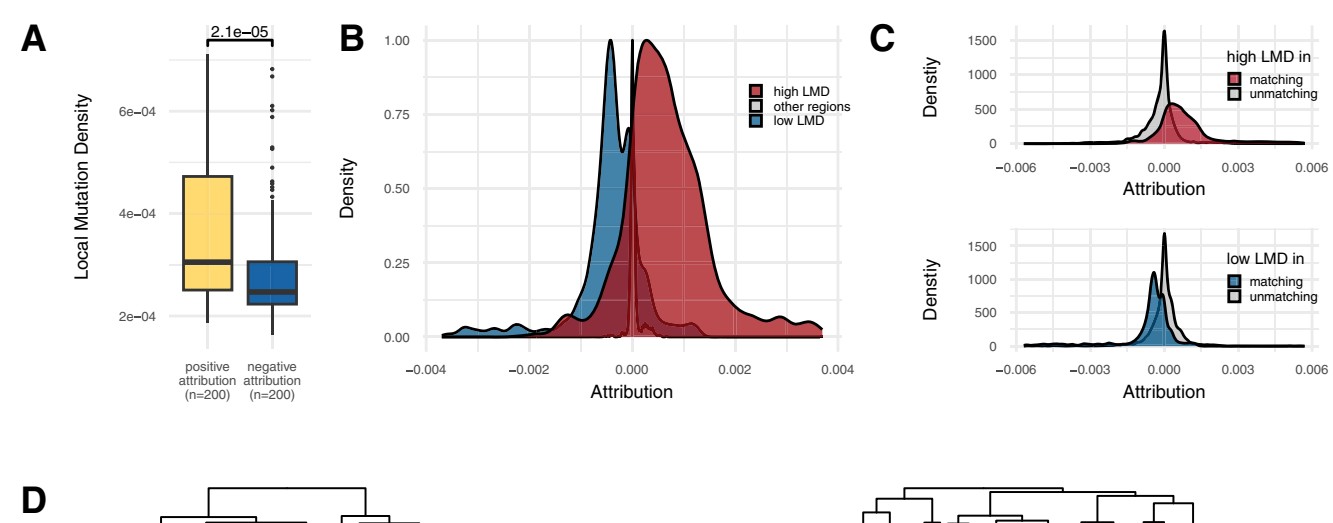

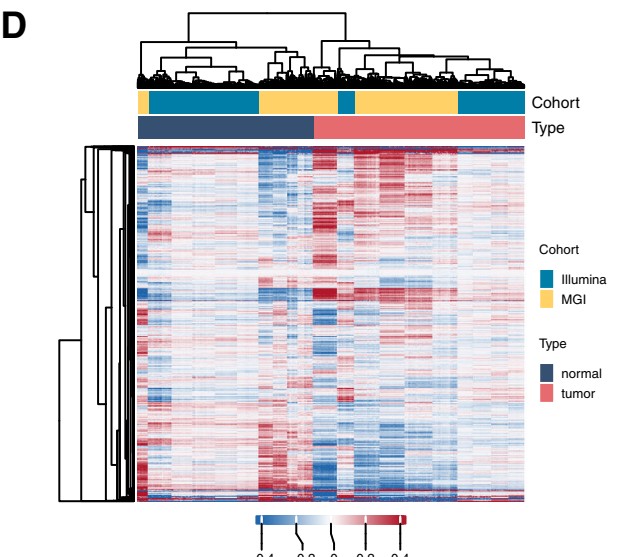

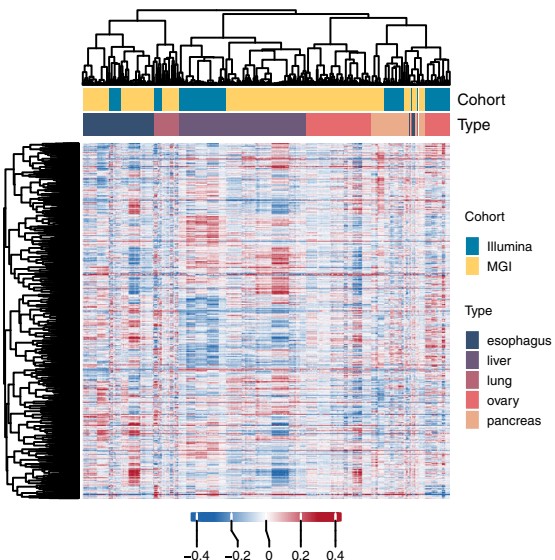

**Fig. 5 | Interpretation of the genome model. A** LMD values obtained from tumor tissues in the LVD regions with positive or negative attribution for cancer samples in the genome model for cancer detection on the MGI training cohort. *P* values from the Wilcoxon test are indicated (two-sided). Each box indicates IQR and median, whiskers indicates 1.5× IQR, black dots indicates outlier. **B** Distribution of the attribution values assigned by the cancer localization genome model of the MGI training cohort to high or low LMD regions in comparison to other regions of the PCAWG cancer type matching the given prediction label. **C** Comparison of the attribution values assigned by the cancer localization genome model to high (upper) or low (lower) LMD regions of the PCAWG cancer types matching versus unmatching with the given prediction label. **D** Clustered heatmaps of normalized attribution values in the genome model for (left) cancer detection and (right) cancer localization across normal samples and patient samples. The cancer types commonly present in the MGI training cohort and Illumina training cohort were included. Only the samples with a test prediction score above the threshold were included. The attribution values of all genome model features were normalized as the z score. Source data are provided as a Source Data file.

regions. The LMD was calculated in 1 Mb running windows across the genome for each sample. Cancer types were defined as the organ of origin. Specific LMD high or low regions were identified using quasi-likelihood F-tests on the trimmed mean of the M-value (TMM)-normalized counts[27]. High or low LMD regions were selected at an FDR of 0.05 and sorted based on the fold change. For each cancer type, the top 25 regions with mutation enrichment and the top 25 regions with mutation depletion were identified as the cancer type-specific high and low LMD regions, respectively.

**Genome model input processing and training**
The cfDNA bam files were processed using the GATK data preprocessing pipeline for variant discovery[25]. The GATK data preprocessing pipeline includes duplicate marking, indel realignment, and base quality score recalibration. SAMtools with the -f 2 -F 2308 -q 30 options was used to filter out supplementary, unmapped, and low mapping quality reads, leaving only properly paired reads. Additionally, the SAMtools view option was used to extract canonical chromosomes.

For variant calling, we ran VarScan opting for at least one variant supporting read, at least 3× depth at a variant position, and an average base quality of at least 30[28]. After variant calling, single nucleotide variants were extracted while filtering out those in the blacklisted regions[29]. The variants present in the cfDNA normal reference panel were removed. Additional germline variants were also removed using the dbSNP, 1000 Genomes, HapMap, ExAC, and gnomAD databases. Additional artifact filtering processes were implemented. First, variants with a variant allele frequency of 100% were eliminated. Second, discordant reads were discarded by checking whether the variant allele was found on only the forward or reverse reads. Third, multi-nucleotide variants were discarded. Finally, variants at the ends of a read were removed considering the possibility of sequencing errors. The read end was defined as 10 bp from the beginning or end of each read.

On the basis of the filtered variants, the LVD was calculated in 1 Mb running windows across the genome for each sample. The number of variants in each bin was divided by the total number of variants in the

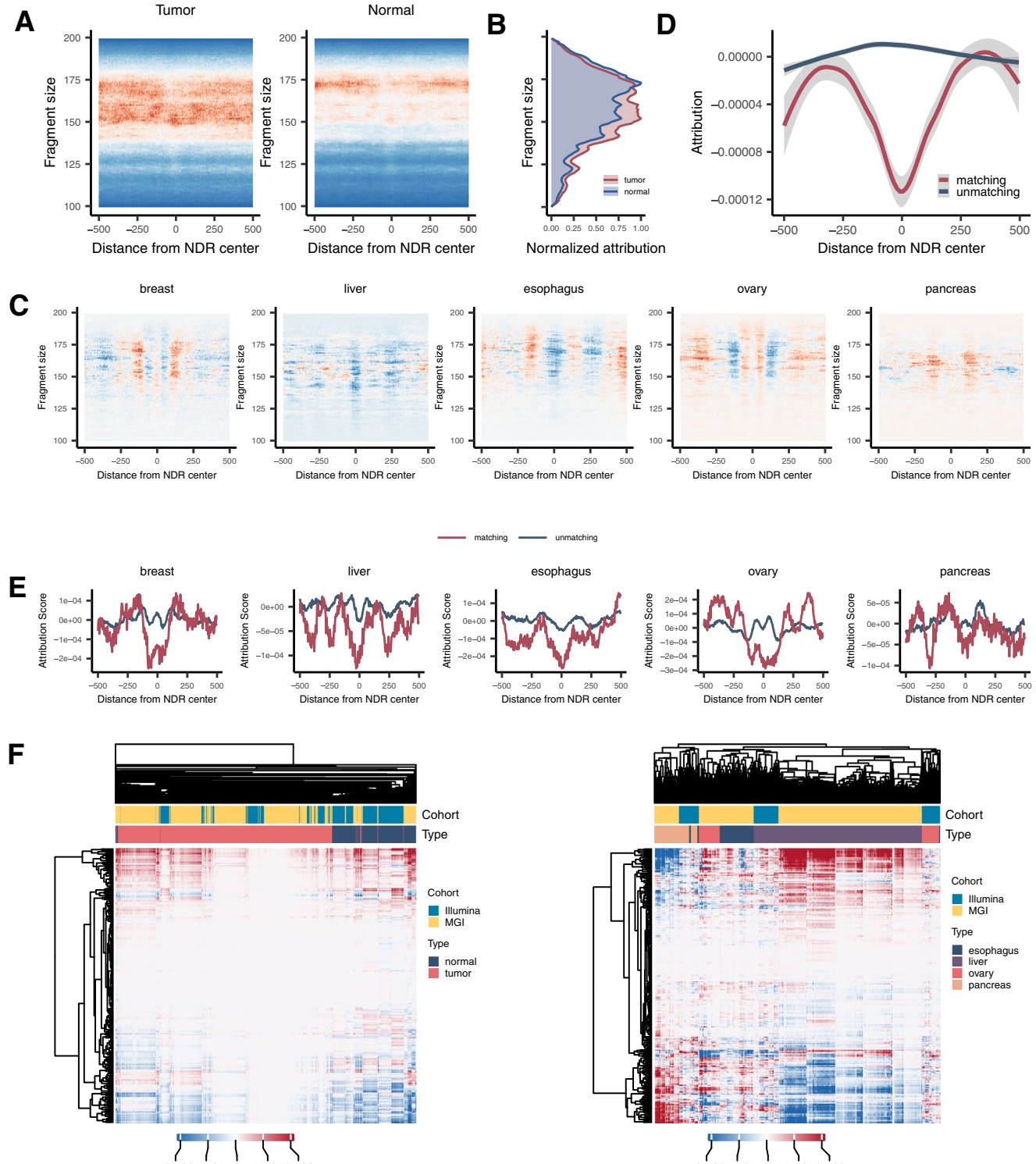

**Fig. 6 | Interpretation of the epigenome model. A** Attribution values mapped to the tissue-specific V-plots of the epigenome model for cancer detection on the MGI training cohort. The average attribution values across the tissue-specific V-plots are compared for cancer samples (left) and normal samples (right). **B** Distribution of the normalized attribution values according to fragment size. **C** Attribution values mapped to the tissue-specific V-plots of the epigenome model for tissue-of-origin localization on the MGI training cohort. The average attribution values across the tissue-specific V-plots are shown for each cancer type. **D** Attribution values of the epigenome model for cancer localization on the MGI training cohort, as averaged across the NDRs of the matching (red) versus unmatching (blue) cancer types, and plotted according to the distance from the NDR midpoints. **E** The same plot as **D** broken down by cancer types. Attribution values were smoothed using lowess regression. **F** Clustered heatmaps of normalized attribution values in the epigenome model for (left) cancer detection and (right) cancer localization across normal samples and patient samples. The cancer types commonly present in the MGI training cohort and Illumina training cohort were included. Only the samples with a test prediction score above the threshold were included. The attribution values of all epigenome model features were normalized as the z score. Source data are provided as a Source Data file.

sample. After discarding regions in which no variant was found in more than 50% of samples, 2,726 genomics regions were left to be used as features of the genome model. For variant-type features, 6 types of single nucleotide substitutions (C>A, C>G, C>T, T>A, T>C, and T>G), 48 types of dinucleotides consisting of each substitution and either the 5′- or 3′-flanking base, and 96 types of trinucleotides consisting of each substitution and both the 5′- or 3′-flanking bases were used. The frequency of each of the 150 variant types was computed by dividing by the total number of variants per sample. In total, 2726 LVD values and 150 variant-type frequencies were used as features for the genome model.

The genome model was constructed using TensorFlow deep neural networks to train the cancer detection and localization models. The genome model consisted of dense layers, batch normalization, activation, and dropout. For the cancer detection model, the output layer employed the sigmoid function and the loss function based on binary cross-entropy. For the tissue-of-origin localization model, the output layer employed the softmax function and the loss function using categorical cross-entropy. Optimization of the genome model was performed by the AdamW optimizer. Because of the imbalanced number of samples, the initial weight of the output layer was modified by calculating the class weight using Scipy[30].

Bayesian optimization was used with 200 trials to find the best hyperparameter set to maximize the performance on the validation set. The list of the hyperparameters is provided in Supplementary Data 10. To prevent overfitting, the early stopping method was applied with a patience of 50. After finding the best hyperparameter set, the genome model was trained 30 times with different random states, and the model with the lowest loss value on the validation set was selected as the final model.

## Processing cell line data to establish the epigenome model pipeline

The GM12878 ATAC-seq data were downloaded from the GEO database, and the SRA file was converted into the fastq format. The K562 ATAC-seq data were downloaded from the ENCODE portal in the fastq format. The ATAC-seq fastq datasets underwent an adapter trimming process using Trim Galore, and were aligned using BWA with the default options. Duplicate marking was performed using GATK4[25], and the duplicate reads were removed using SAMtools. Supplementary and low mapping quality reads were removed using SAMtools with the -f 2 -F 2048 -q 30 options, leaving only properly paired reads. Peak calling was performed for the filtered ATAC-seq reads using HMMRATAC[21] with the default options or MACS2[22] with the -shift −75 -extsize 150 -nomodel -nolambda -call-summits -q 0.05 -B −SPMR options. In total, 10,9360 peaks and 24,937 peaks were called by MACS2[22] and HMMRATAC[21], respectively, in GM12878, and 17,5733 peaks and 35,708 peaks were called by MACS2[22] and HMMRATAC[21], respectively, in K562. To identify differential NDRs between the GM12878 and K562 cell lines, the HMMRATAC[21] peaks were intersected to find non-overlapping peaks. A total of 13,132 peaks were called as GM12878-specific NDRs, and 23,515 peaks were called as K562-specific NDRs.

The ChIP-seq and MNase-seq data in the GM12878 and K562 cell lines were downloaded from the ENCODE data portal in bam format. To remove duplicates, supplementary reads, and low mapping quality reads, the same filtering process using GATK4[25] and SAMtools was performed as described above. The filtered ChIP-seq and MNase-seq reads were extended to 150 bp. To refine nucleosome positioning, the MNase-seq reads were resized to 75 bp from the midpoint of the 150 bp read.

## Identification of tissue-specific NDRs

The ATAC-seq data for tumor and normal tissues or PBMCs were obtained from TCGA, ENCODE, and GEO databases. From TCGA portal, 410 samples of 23 cancer types (ACC, BLCA, BRCA, CESC, CHOL, COAD, ESCA, GBM, HNSC, KIRC, KIRP, LGG, LIHC, LUAD, LUSC, MESO,

PCPG, PRAD, SKCM, STAD, TGCT, THCA, and UCEC) were downloaded. Because TCGA ATAC-seq data were aligned to the hg38 genome assembly, TCGA bam files were converted into the fastq format and realigned to hg19 using BWA-MEM. Filtering duplicates, supplementary reads, and low mapping quality (<30) reads were carried out as tested using the cell line data. Using the filtered ATAC-seq reads, the NDR peaks were called using HMMRATAC[21]. Samples with less than 30,000 NDRs were excluded, leaving 266 of 410 samples for further analyses. The CESC cancer type was excluded because we were left with only one sample. From the ENCODE project, 17 ATAC-seq bam files for two normal tissues (pancreas and ovary) were downloaded. The NDR peaks were called using HMMRATAC[21]. Among the 17 samples, 15 samples with more than 30,000 NDRs were left for further analyses. In addition, 26 ATAC-seq bam files for five PBMC types (CD4 T cells, CD8 T cells, B cells, monocytes, and NK cells) were downloaded from the GEO database. The bam files were merged according to cell types to compensate for low sequencing depths. After merging the bam files, the NDR peaks were called using HMMRATAC[21].

Following the peak processing and filtering pipeline, tissue-specific NDRs were identified by using the 25 filtered samples. The NDR peaks called from each sample were merged, and peaks with the strongest signal were selected among overlapping peaks, resulting in 61,5401 meta-peaks. The ATAC-seq reads that were shorter than 150 bp and overlapped with the midpoint of the meta-peaks were counted to construct a read count matrix. The matrix was normalized using the TMM function of edgeR[27] and subjected to the limma-voom algorithm[31] of edgeR for the selection of the 10,000 tissue-specific NDRs with the lowest P value. NDR-wise min-max normalization[30] was applied for visualization of the tissue specificity of the selected NDRs by a heatmap.

## Epigenome model input processing and training

The process of filtering duplicate, supplementary, low mapping quality, and unpaired cfDNA fragments was performed as tested using the cell line data. Each fragment was cut in half centering on its midpoint in a similar manner that MNase-seq reads are processed for nucleosome positioning analysis. For each tissue type, the cfDNA fragments were aligned to the identified tissue-specific 10,000 NDRs, and a two-dimensional V-plot was constructed to plot fragment size as a function of the distance from the NDRs. The y-axis representing fragment size contained 100 1-bp bins ranging from 100 bp to 200 bp. The x-axis, representing the distance from the NDRs, was made up of 250 4-bp bins covering ±1000 bp from the NDR midpoint. Finally, the two-dimensional V-plot for each tissue type was added to the channel axis to create a three-dimensional (100, 250, 25) V-plot image as input to the epigenome model. To correct for read depth bias, min-max normalization[30] was applied to the vector of individual pixels of the V-plot image for each sample.

The TensorFlow CNN architecture was employed for the development of the epigenome model. The epigenome model was designed with two convolutional layers and one fully connected layer. On the first convolutional layer, a normalized three-dimensional V-plot image was convoluted with 250 kernels, followed by batch normalization, ReLU activation, and dropout. The second convolution layer employed one kernel for convolution followed by batch normalization, ReLU activation, and dropout. Following the convolution steps, the flattened nodes were connected to fully connected layers. For the cancer detection model, the output layer was composed of one node with the sigmoid function, and binary cross-entropy was used for the loss function. For the tissue-of-origin localization model, the output layer was composed of the number of tumor sites with the softmax function, and categorical cross-entropy was used for the loss function. Because of the imbalanced number of samples, the initial weight of the output layer was modified by calculating the class weight using Scipy[30].

Bayesian optimization was used with 200 trials to find the best hyperparameter set to maximize the performance on the validation set. The list of the hyperparameters is provided in Supplementary Data 10. To prevent overfitting, the early stopping method was applied with a patience of 50. After finding the best hyperparameter set, the epigenome model was trained 30 times with different random states, and the model with the lowest loss value on the validation set was selected as the final model.

## Cnv, fragpattern, and fragsize model input processing and training

The whole genome was split into non-overlapping 5 Mb windows. To filter out the blacklisted regions including telomeres and centromeres, the windows containing N nucleotides or with low GC contents were discarded. To construct the cnv model, read coverage at each bin was obtained and normalized using the z score. The input data for the fragpattern model were processed, as previously described[9]. Briefly, locally weighted scatterplot smoothing (LOWESS) regression analysis was separately applied for short (100–150 bp) and long (151–220 bp) fragments with a span setting of 0.75. The LOWESS estimate of coverage was subtracted from the original measurement to separately obtain residuals for the short and long fragments. The genome-wide median short and long estimates of coverage were added back. The corrected values of total and short fragment coverage were z-score normalized across all bins for each sample. For the input of the fragsize model[8], cfDNA fragments were broken down by length, increasing by 2 bp from 0 to 250 bp. The 125 features were normalized using the z score.

For training of the cnv, fragpattern, and fragsize models, gradient boosting decision trees were implemented by XGBoost[32]. Hyperparameters, including n_estimators, learning_rate, max_depth, min_child_weight, and colsample_bytree, were searched using a random search method with three-fold cross-validation. The XGBoost model was fitted using the log loss evaluation metrics, and the lowest loss model on the validation set was selected as the final model. The binary:logistic objective function was used for cancer detection, and the multi:softmax objective function was used for tissue-of-origin localization.

## Processing of the DELFI cohort data

We used the DELFI dataset with 1-2× cfDNA WGS of 214 healthy samples and 208 cancer patients to validate our algorithm. Cancer patient samples include breast ($n = 54$), pancreatic ($n = 34$), ovarian ($n = 28$), colorectal ($n = 27$), gastric ($n = 27$), lung ($n = 12$), and bile duct cancer ($n = 26$). Following the approval of their Data Access Committee (DAC), duplicate marked bam files of the DELFI dataset were obtained from European Genome-Phenome Archive (EGA). Genome, epigenome, cnv, fragpattern, and fragsize input features were processed using duplicate marked bam as described in the sections "Genome model input processing and training", "Epigenome model input processing and training" and "Cnv, fragpattern, and fragsize model input processing and training".

## Model training using the training cohort

Each training cohort (MGI, Illumina and DELFI cohort) for the MGI and Illumina sequencing platforms was partitioned into five groups for the application of the stratified five-fold cross-validation. At each iteration, four groups in the training set were further divided into three training sets and one validation set. Using this method, all samples were given a test prediction score from each model. Using the test prediction score of all training cohort samples, we calculated the ROC-AUC score for cancer detection and the accuracy score for tissue-of-origin localization. The confidence interval for sensitivity was calculated from 1000 bootstrap samplings with replicates at 95%, 98%, and 99% specificity. The cancer localization model was developed by using either all cancer samples in the training cohorts or the cancer samples correctly identified by the combined cancer detection model with 98% specificity.

The number of correctly predicted samples was 1188 out of 1359 for the MGI cohort and 644 out of 940 samples for the Illumina cohort.

## Model prediction using the validation cohort

To validate the robustness of the models, validation cohorts were generated using independent batches from the training cohorts. The combined, genome, epigenome, cnv, fragpattern, and fragsize algorithms were used to predict validation cohorts. The average of five prediction values was used as the final prediction score. Using the final prediction score, we calculated the ROC-AUC score. Also, the confidence interval for sensitivity was calculated from 1000 bootstrap samplings with replicates at 95% specificity.

## Model interpretation using integrated gradients

The attribution score, a per-feature score based on the feature's contribution to the model's output when making a prediction, was calculated using an integrated gradient method[33] that considers all gradients from the baseline input to the real input. TensorFlow was used to implement the integrated gradient method. To calculate the attribution scores, a zero array was used as the baseline input. From the baseline to each input array, 50 interpolated arrays were generated. The gradients of each interpolated array were obtained and averaged to calculate the attribution scores.

From the cancer detection genome model, cancer samples with a test prediction score > 0.8 and normal control samples with a test prediction score <0.2 were selected from the MGI training cohort and Illumina training cohort, respectively. As a result, 601 cancer samples and 306 normal control samples from the MGI training cohort and 391 cancer samples and 519 normal samples from the Illumina training cohort were used for interpretation analysis. The attribution score was calculated as described above. Positive attribution features of cancer were defined as top 100 regions with positive attribution value in cancer patient predictions and negative attribution value in normal sample predictions. Also, negative attribution features of cancer were defined as top 100 regions with negative attribution value in cancer patient predictions and positive attribution value in normal sample predictions. We tested whether positive attribution features of cancer actually had high LMD values in the PCAWG tissue data, and negative attribution features of cancer had low LMD values in the PCAWG tissue data.

For the genome model for cancer localization, cancer samples whose localization was predicted correctly with a test prediction score > 0.6 were selected from the MGI training cohort and illumina training cohort, respectively, for interpretation analysis. As a result, 640 breast cancer, 166 liver cancer, 130 esophageal cancer, 134 ovary cancer, 59 pancreatic cancer, and 34 lung cancer cfDNA samples from the MGI training cohort were selected. 99 liver cancer, 25 esophageal cancer, 53 ovary cancer, 41 pancreatic cancer, and 20 colon cancer were selected from the Illumina training cohort. The attribution score was calculated as described above. To evaluate the attribution of the reference LMD values obtained from the PCAWG data, the sample-wise attribution scores were compared among the high LMD regions, low LMD regions, and other LMD regions of each cancer type. We tested whether high LMD regions had positive attribution whereas low LMD regions had negative attribution, specifically for the PCAWG cancer type matching with the given cfDNA sample but not for the other PCAWG cancer types unmatching with the given cfDNA sample.

From the cancer detection epigenome model, cancer samples with a test prediction score >0.8 and normal control samples with a test prediction score <0.2 were selected from the MGI training cohort and Illumina training cohort, respectively, for interpretation analysis. As a result, 1152 cancer samples and 164 normal control samples from the MGI training cohort were selected. A total of 312 cancer samples and 401 normal control samples from the Illumina training cohort were selected. To display attribution values as 2D image, the absolute value

of the attribution array (sample size, 100, 250, 25) was averaged along the axes 0 and 2 for the normal control and cancer patients, respectively. To examine the effect of fragment size, the absolute value of the attribution array (sample size, 100, 250, 25) was averaged along the axes 0, 2, and 3 separately for the normal control and cancer samples, respectively. Min-max normalization[30] was applied across fragment size to compare its distribution between the normal control and cancer samples. To draw a heatmap, we used normalized attribution values of fragment length and NDR position. Normalized attribution values of the fragment length were processed by averaging the attribution arrays (sample sizes, 100, 250, 25) along axes 2 and 3, and normalized attribution values of the NDR position were processed by averaging the attribution arrays (sample sizes, 100, 250, 25) along axes 1 and 3.

For the epigenome model for cancer localization, cancer samples whose localization was predicted correctly were selected from the MGI training cohort and Illumina training cohort, respectively. As a result, 591 breast cancer, 155 liver cancer, 71 esophageal cancer, 109 ovary cancer, 26 pancreatic cancer, 3 lung cancer samples from the MGI training cohort and 117 liver cancer, 20 esophageal cancer, 81 ovary cancer, 79 pancreatic cancer from the Illumina training cohort were used for interpretation analysis. To display attribution values as 2D image, the attribution array (sample size, 100, 250, 25) was averaged along the axes 0 and 2 for each cancer type, respectively. To examine the effect of genomic distances from NDRs, the tissue-specific NDRs were classified into NDRs of cancer types matching the given cfDNA sample and those of cancer types unmatched with the given cfDNA sample. The attribution array (sample size, 100, 250, 1) for the matched NDRs and the attribution array (sample size, 100, 250, 24) for the unmatched NDRs were averaged along the axes 0, 1, and 3 within each cancer type. To draw a heatmap, we used normalized attribution values of fragment length and NDR position. Normalized attribution values of the fragment length were processed by averaging the attribution arrays (sample sizes, 100, 250, 25) along axes 2 and 3, and normalized attribution values of the NDR position were processed by averaging the attribution arrays (sample sizes, 100, 250, 25) along axes 1 and 3.

### Tumor fraction estimation

Tumor fraction in cfDNA was estimated by using ichorCNA[3]. The reads with mapping quality > 30 reads were selected to compute read coverage across 1 Mb non-overlapping windows using the readCounter function in the HMMcopy R package[3]. Next, 417 normal control samples in the MGI cohort and 728 normal control samples in the Illumina cohort were used to create a normal reference dataset for each platform. Tumor fraction was estimated using the ichorCNA algorithm with the platform-specific normal reference dataset.

### Downsampling analysis

The average depth of cfDNA WGS was calculated to perform downsampling. SAMtools depth was used to calculate the average depth of cfDNA WGS. To make the desired depth of cfDNA WGS, required downsampling ratio was calculated by dividing the desired depth by the actual depth. Downsampling was performed using SAMtools view with the -b -h and -s [downsampling ratio] options. Genome and epigenome input features were processed using downsampled bam as described in the sections "Genome model input processing and training" and "Epigenome model input processing and training".

### Statistics and reproducibility

We performed wilcoxon two-sided test for statistical analysis. For study design, we did not use statistical method to predetermine sample size. Also no data were excluded from the analyses. The experiments were not randomized and the investigators were not blinded to allocation during experiments and outcome assessment.

### Reporting summary

Further information on research design is available in the Nature Portfolio Reporting Summary linked to this article.

## Data availability

The 3366 cfDNA WGS data generated in this study have been deposited in the European Genome-phenome Archive (EGA) [https://ega-archive.org/datasets/EGAD00001009335]. Our data will be made available under approval by the data access committee. There are no restrictions on who will be granted access to this data. Access will be provided within approximately one month and be available for one year. DELFI cfDNA WGS data also can be obtained from EGA [https://ega-archive.org/datasets/EGAD00001005339]. Tumor somatic mutation MAF data were downloaded from PCAWG [https://dcc.icgc.org/releases/PCAWG/consensus_snv_indel/final_consensus_passonly.snv_mnv_indel.tcga.controlled.maf.gz, https://dcc.icgc.org/releases/PCAWG/consensus_snv_indel/final_consensus_passonly.snv_mnv_indel.icgc.controlled.maf.gz]. Tissue ATAC-seq bam files were downloaded from TCGA [https://portal.gdc.cancer.gov/]. ATAC-seq of GM12878 and K562 were downloaded from SRA [GM128782: https://www.ncbi.nlm.nih.gov/sra/?term=SRR891268, K562: https://www.ncbi.nlm.nih.gov/sra/?term=SRR8137174]. MNase-seq and histone modification ChIP-seq data were downloaded from ENCODE [GM12878 MNase-seq: https://www.encodeproject.org/files/ENCFF000VLH/@@download/ENCFF000VLH.bam, K562 MNase-seq: https://www.encodeproject.org/files/ENCFF000VMJ/@@download/ENCFF000VMJ.bam, GM12878 H3K27ac: https://www.encodeproject.org/files/ENCFF197QHX/@@download/ENCFF197QHX.bam, GM12878 H3K9ac: https://www.encodeproject.org/files/ENCFF415YCS/@@download/ENCFF415YCS.bam, GM12878 H3K4me1: https://www.encodeproject.org/files/ENCFF753GZX/@@download/ENCFF753GZX.bam, GM12878 H3K4me2: https://www.encodeproject.org/files/ENCFF794KPF/@@download/ENCFF794KPF.bam, GM12878 H3K4me3: https://www.encodeproject.org/files/ENCFF375WTP/@@download/ENCFF375WTP.bam]. Source data are provided with this paper.

## Code availability

Code for training and prediction of genome, epigenome model can be accessed at https://github.com/kaistomics/cfWGS.

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

## Acknowledgements

This research was supported by the National Research Foundation (NRF) funded by the Ministry of Science and ICT (NRF-2017M3A9A7050612 and 2020M2D9A3094213).

## Author contributions

M.B., G.K., J.K.C. conceived the study, designed all the experiments, and wrote the manuscript. M.B., G.K. developed genome and epigenome model and performed all analysis. H.P helped analyze and draw figures. T.R.L., J.M.A., J.L., D.K., C.S.K., E.H.C. performed cfDNA sequencing and basic alignment. S.R.P., K.B.S., E.J., D.O., J.W.L., Y.S.P., K.W.S., J.S.B., B.H.K., J.H.S., M.H.K., G.M.K., E.K.C., H.C.K., S.Y.K., S.M.W., J.E.L., J.M.R. collected samples and summarized clinical information. J.K.C. and E.H.C. supervised the study.

## Competing interests

The authors declare no competing interests.

## Additional information

[1]Department of Bio and Brain Engineering, KAIST, Daejeon, Republic of Korea. [2]Genome Research Center, GC Genome, Yongin, Republic of Korea. [3]Department of Oncology, Asan Medical Center, University of Ulsan College of Medicine, Seoul, Republic of Korea. [4]Division of Hepato-Biliary and Pancreatic Surgery, Department of Surgery, Asan Medical Center, University of Ulsan College of Medicine, Seoul, Republic of Korea. [5]Department of Radiation Oncology, Samsung Medical Center, Sungkyunkwan University School of Medicine, Seoul, Republic of Korea. [6]Department of Obstetrics and Gynecology, Samsung Medical Center, Sungkyunkwan University School of Medicine, Seoul, Republic of Korea. [7]Division of Pulmonary and Critical Care Medicine, Department of Internal Medicine, Seoul National University Hospital, Seoul, Republic of Korea. [8]Division of Hepatopancreatobiliary Surgery and Liver Transplantation, Department of Surgery, Asan Medical Center, University of Ulsan College of Medicine, Seoul, Republic of Korea. [9]Department of Gastroenterology, Asan Medical Center, University of Ulsan College of Medicine, Seoul, Republic of Korea. [10]Center for Liver and Pancreatobiliary Cancer, National Cancer Center, Goyang, Republic of Korea. [11]Division of Medical Oncology, Department of Internal Medicine, Yonsei Cancer Center, Yonsei University College of Medicine, Seoul, Republic of Korea. [12]AIMA, Inc., Avison Biomedical Research Center, Seoul, Republic of Korea. [13]Department of Radiation Oncology, Seoul National University College of Medicine, Seoul, Republic of Korea. [14]Department of Laboratory Medicine, National Cancer Center, Goyang, Republic of Korea. [15]Department of Surgery, Samsung Medical Center, Seoul, Republic of Korea. [16]These authors contributed equally: Mingyun Bae, Gyuhee Kim. ✉e-mail: ehcho@gccorp.com; jungkyoon@kaist.ac.kr

