## [Peer Review File · Nature Communications]

Integrative modeling of tumor genomes and epigenomes for enhanced cancer diagnosis by cell-free DNAREVIEWER COMMENTS

Reviewer #1 (Remarks to the Author): expert in convolution neural networks and machine learning

Major Concerns:

1. Not sure it should be called Integrative modeling, since it is only calculating the average score from independent genome model and epigenome model.
2. How the DELFI used samples were processed is not mentioned anywhere in the manuscript.
3. The high sensitivity and specificity are impressive. But here are some model evaluation-related concerns:

In last paragraph of Page.20, author mentioned that the best performance from 30 times running was selected as the validation performance, but it will be important for the readers to know the variance of the 30 models' performance to determine the robustness of the model.

4. It is not clear how results shown in Supplementary figure 4 was derived. How the evaluation was done on DELFI dataset is not mentioned in methods part. e.g., with so high a sensitivity and specificity, is it cross-validation results (meaning model trained on DELFI samples again) or just test results predicted using MGI/Illumina-trained model? If the latter, which platform-trained models were used to predict DELFI samples? It is critical results determining the model performance, and should be described to help readers understanding the whole evaluation process.

5. It would be a more comprehensive evaluation if the MGI-trained model can be applied in Illumina samples.

6. In model interpretation, the consistence between MGI-trained and Illumina-trained model feature importance could to be evaluated. This is helpful in determining whether models are overfitting their own datasets, or discovered real early cancer genome and epigenome changes.

Reviewer #3 (Remarks to the Author): expert in cfDNA genomics

In this paper, the authors proposed a work that tried to diagnose cancer through cfDNA WGS data by integrating both genome and epigenome features. The data size and performance of this work is impressive, but there are some key issues that the authors didn't address well. I have some concerns about the significance and methods of this work.

(1) The key significance of the work seems to be the algorithm that integrate both genome and epigenome features. However, there are few details about the algorithm. For example, no figure about the algorithm was shown in the article, which put obstacles to the readers to understand the algorithm. A pipeline should be provided to assess the whole algorithm process, e.g. the detail of CNN and the usage of different data.

Actually, it is intuitional to merge mutation and fragment information of cfDNA to enhance the accuracy of diagnosis, and there are many works to deal with the issues. How and why the authors develop their algorithm? What's the difference between their algorithm and other works that deal with genomes, epigenomes as well as integration of them? What can other researchers learn from this work to deal with their own problems? I think the authors should explain these issues carefully in both introduction and discussion, which could benefit more to the scientific community.

(2) Understanding the mechanism of deep learning models is essential to trust the models and take advantage from the work for promoting related research. One possible approach is to use some interpretive methods for neural networks; another is to do some extended simulations. For example, what features in V-plot matter more in the diagnosis of cancer? Does all LMDs / NDRs contribute to the diagnosis, or important signals can be enriched into some specific gene sets? I suggest the authors to do more analysis to help the readers understand the significance of the algorithm.

(3) The work used both MGI and Illumina platforms to sequence cfDNA. Can models trained on one platform be used to test data derived from another platform?

(4) The cost of sequencing is an important issue of cfDNA cancer diagnosis. The data derived from the MGI platform are around 5X, which seems to be a costly depth. I suggest the authors to do some downsampling to investigate the relationship between sequencing depth and precision.

(5) The usage of K562 and GM12878 cell line data seems to be unclear. What's the relationship between the data and the downstream analysis?

(6) The authors used five-fold cross-validation to train and test the model. Though the approach seems to be reasonable, I am still curious about why the authors don't used a more straightforward approach to train, validate and test the model with such a large dataset. The authors seem to train a lot parallel models in the whole approach, but which model was used to test the external DELFI dataset? Does the choice of hyper-parameters matter a lot to the final results? If I were a user of the algorithm, how should I merge these parallel models? I suggest the authors to explain the training and testing approach more systematically to help the readers understand the work better and guarantee that there is no information leakage in the approach.

Reviewer #4 (Remarks to the Author): expertise in multi-omics integration

In this study, the authors performed cell-free DNA (cfDNA) whole genome sequencing to generate two test datasets including 2543 patient samples from nine different cancer types and 1241 normal control samples, and a reference dataset for background variant filtering based on 20529 samples from low-depth healthy subjects. An external cfDNA data set consisting of 208 cancer samples and 214 normal controls was also used for additional evaluation. The algorithm incorporates as model criteria the distribution of mutations in tumor tissue and cancer type-specific profiles of chromatin tissue, achieving very high accuracy in cancer detection and tissue origin localization. This integrated model was able to detect early-stage cancers, including pancreatic cancer, with high sensitivity comparable to late-stage cancers. In addition, interpretation of the model revealed the contribution of genomic and epigenomic features to different types of cancer. This methodology could lay the groundwork for accurate cfDNA-based cancer diagnosis, especially at early stages.

The authors are working on an important research topic and have obtained some interesting results. On the other hand, there are several problems, and I recommend a major revision of the manuscripts, paying attention to the following points.

1. Despite the complexity of the research, the number of main figures is small and the text in each figure is small, making it difficult to grasp the overall picture of the research. It would be better to increase the number of main figures and make the letters in the figures larger.
2. Deep neural networks were used in this study, but it is difficult to understand the details in the current version. The details should be shown in Fig. 1 or elsewhere, including the structure of the algorithm.
3. The most serious problem with this article is that, despite the fact that it is a medical and clinical-oriented study, it provides no information about the research institution or the institution from which the clinical specimens were taken. The authors are not even sure if they are cancer experts to begin with, although they have done some clinical-leaning research on cancer in particular. Given the above, it is also impossible to determine whether the results of the study have been properly interpreted. To be honest, the limited information on the research group in this paper makes it difficult to evaluate the article's content. When publishing such clinically oriented papers, it should be possible to determine the type of professional (clinician, medical researcher, informatics researcher) from the peer review stage.
4. Related to 3, the current manuscript is unclear in which region (country) the study was conducted. From the MATERIALS AND METHODS, I have determined that it is probably a Chinese research group, but in that case, the genetic background by ethnic group should also be noted. A number of studies have reported results indicating that genetic backgrounds in Western countries differ significantly from those in Asian regions to begin with. A method of analysis that mixes up genetic backgrounds, as in this case, may lead to misinterpretation of the results.
5. To be honest, my first impression from reading the article is that the AUC values in Figure 2 are too high, which may have caused over fitting. It is difficult to determine accuracy from retrospective studies alone, and it would be impossible to properly determine usefulness without

conducting prospective studies.

6. In the judgment of cancer experts, the molecular mechanisms of cancer development differ greatly in different organs. It is difficult to determine whether the method of training all nine types of cancer in a jumble as in this case is really useful. In particular, in Figures 2E and F, the colon, biliary tract, and head and neck data are almost statistically meaningless because the number of samples is small and the error bars are too large. I do not understand the significance of presenting such data. Similarly, the data in NA in Figure 2E have too few samples and very large error bars. I find the data to be completely meaningless from a scientific standpoint.

RESPONSE TO REVIEWERS' COMMENTS

Reviewer #1

Reviewer #1 – Major comments:

1. Not sure it should be called integrative modeling, since it is only calculating the average score from independent genome model and epigenome model.

Our apologies for the confusion. By integrative modeling, we referred to our approach that combines various public data and our own datasets (normal cfDNA reference panel consisting of 20,529 samples, PCAWG, TCGA, ENCODE, GEO, dbSNP, 1000 Genomes, HapMap, ExAC, and gnomAD). Especially, modeling of the tumor tissue mutation and chromatin profiles from PCAWG and TCGA played a critical role in developing our prediction models. Our normal reference panel also played an essential role in filtering artifactual variants in developing the genome model. For clarification, we have modified the Figure 1 as follows.

[Pipeline of integrative modeling with reference datasets]

2. How the DELFI used samples were processed is not mentioned anywhere in the manuscript.

We apologize for the lack of detailed information. We have now added the description of DELFI cohort processing procedures in the method section as follows (page 34, line 11):

Processing of the DELFI cohort data

We used the DELFI dataset with 1-2x cfDNA WGS of 214 healthy samples and 208 cancer patients to validate our algorithm. Cancer patient samples include breast (n=54), pancreatic (n=34), ovarian (n=28), colorectal (n=27), gastric (n=27), lung (n=12), and bile duct cancer (n=26). Following the approval of their Data Access Committee (DAC), duplicate marked bam files of the DELFI dataset were obtained from European Genome-Phenome Archive (EGA). Genome, epigenome, cnv, fragpattern, and fragsize input features were processed using duplicate marked bam as described in the sections "Genome model input processing and training", "Epigenome model input processing and training" and "Cnv, fragpattern, and fragsize model input processing and training".

3. The high sensitivity and specificity are impressive. But here are some model evaluation-related concerns: In last paragraph of Page.20, author mentioned that the best performance from 30 times running was selected as the validation performance, but it will be important for the readers to know the variance of the 30 models' performance to determine the robustness of the model.

We thank the reviewer for this constructive comment giving us an opportunity to prove the robustness of our models. To address this comment, we examined the variance of the 30 models' performance in terms of the loss and AUC for the genome and epigenome model in cancer detection and tissue-of-origin localization as follows. These results are provided in Supplementary Figures 4 and 5 (for cancer detection and localization, respectively). Although the initial seed was changed during the 30 repetitions, no significant difference in performance was observed.

[Performance variation in cancer detection]

A: Genome model on MGI training cohort

B: Epigenome model on MGI training cohort

C: Genome model on Illumina training cohort

D: Epigenome model on Illumina training cohort

[Performance variation in tissue-of-origin localization]

- A: Genome model on MGI training cohort
- B: Epigenome model on MGI training cohort
- C: Genome model on Illumina training cohort
- D: Epigenome model on Illumina training cohort

4. It is not clear how results shown in Supplementary figure 4 was derived. How the evaluation was done on DELFI dataset is not mentioned in methods part. e.g., with so high a sensitivity and specificity, is it cross-validation results (meaning model trained on DELFI samples again) or just test results predicted using MGI/Illumina-trained model? If the latter, which platform-trained models were used to predict DELFI samples? It is critical results determining the model performance, and should be described to help readers understanding the whole evaluation process.

Our apologies for the confusion. In the original Supplementary Figure 4 (currently Supplementary Figure 6), the DELFI cohort data was used for model training. By going through the same training and cross-validation processes, we were able to fairly compare the performance of our models with the DELFI algorithm itself. The details of these processes involving the DELFI data are provided in Supplementary Figure 3A of the revised manuscript as follows (page 34, line 24):

Model training using the training cohort

Each training cohort (MGI, Illumina, and DELFI cohort) for the MGI and Illumina sequencing

platforms was partitioned into five groups for the application of the stratified five-fold cross-validation. At each iteration, four groups in the training set were further divided into three training sets and one validation set. Using this method, all samples were given a test prediction score from each model. Using the test prediction score of all training cohort samples, we calculated the ROC-AUC score for cancer detection and the accuracy score for tissue-of-origin localization. The confidence interval for sensitivity was calculated from 1,000 bootstrap samplings with replicates at 95%, 98%, and 99% specificity. The cancer localization model was developed by using either all cancer samples in the training cohorts or the cancer samples correctly identified by the combined cancer detection model with 98% specificity. The number of correctly predicted samples was 1,188 out of 1,359 for the MGI cohort and 644 out of 940 samples for the Illumina cohort.

During the revision, we also used the DELFI data as the test dataset for our MGI/Illumina-trained models. The results are described in our response to comment #5 below.

5. It would be a more comprehensive evaluation if the MGI-trained model can be applied in Illumina samples.

We appreciate this critical comment. To address this point comprehensively, we performed external validation by applying the combined model trained on one cohort to the training and validation dataset of the other cohorts. When the MGI-trained model was evaluated using the Illumina training cohort, Illumina validation cohort, and DELFI cohort, the performance was an ROC-AUC of 0.90, 0.88, and 0.83, respectively (Supplementary Figure 10C; attached below). When the Illumina-trained model was evaluated using the MGI training cohort, MGI validation cohort, and DELFI cohort, the performance was 0.84, 0.82, and 0.90, respectively (Supplementary Figure 10C; attached below). Additionally, the normal and tumor samples of the MGI cohort were clearly segregated by the prediction scores trained with the Illumina cohort, and *vice versa* (Supplementary Figure 10D; attached below).

[Performance of external validation]

6. In model interpretation, the consistence between MGI-trained and Illumina-trained model feature importance could to be evaluated. This is helpful in determining whether models are overfitting their own datasets, or discovered real early cancer genome and epigenome changes.

We appreciate this constructive comment. To address this point systematically, we performed hierarchical clustering of samples (columns) and features (rows) on the basis of feature attribution values. For both the MGI-trained and Illumina-trained model, we observe that the samples are separated not by the cohort (Illumina, MGI, or DELFI) but by the sample type (tumor or normal) (Figure 5D and Figure 6F). The same analysis was performed also by including the DELFI data, the results of which are attached below. The results including the DELFI data were not included in the manuscript because we were left with only two common tumor types by including the DELFI cohort.

[Clustering by feature attribution from cancer detection by the genome model (left) and epigenome model (right)]

[Clustering by feature attribution from cancer localization by the genome model (left) and epigenome model (right)]

Reviewer #3

Reviewer #3 – General comments:

In this paper, the authors proposed a work that tried to diagnose cancer through cfDNA WGS data by integrating both genome and epigenome features. The data size and performance of this work is impressive, but there are some key issues that the authors didn't address well. I have some concerns about the significance and methods of this work.

We thank the reviewer for the detailed comments and for acknowledging the importance of the subject matter. We have made efforts to address the points raised by the reviewer, especially by describing model interpretation more comprehensively, and now feel that our paper has been improved considerably thanks to the constructive comments.

Reviewer #3 – Major comments:

1. The key significance of the work seems to be the algorithm that integrate both genome and epigenome features. However, there are few details about the algorithm. For example, no figure about the algorithm was shown in the article, which put obstacles to the readers to understand the algorithm. A pipeline should be provided to assess the whole algorithm process, e.g. the detail of CNN and the usage of different data. Actually, it is intuitional to merge mutation and fragment information of cfDNA to enhance the accuracy of diagnosis, and there are many works to deal with the issues. How and why the authors develop their algorithm? What's the difference between their algorithm and other works that deal with genomes, epigenomes as well as integration of them? What can other researchers learn from this work to deal with their own problems? I think the authors should explain these issues carefully in both introduction and discussion, which could benefit more to the scientific community.

We apologize for the lack of detailed information. We now have provided further clarifications on model building procedures, model structures, and learning hyperparameters in the form of schematic illustrations (Figure 1 and Supplementary Figure 3A; attached below).

[Schematic of model building procedures (Figure 1)]

[Schematic of training processes and structures (Supplementary Figure 3A)]

Although some previous studies also used mutation and fragmentation features, our study significantly improved the performance of cancer diagnosis by modeling of large-scale tumor tissue data. As for the genome model using variant features, most previous studies focused on driver mutations of oncogenes and tumor suppressor genes such as TP53 and KRAS. In contrast, we were able to also include passenger mutations in our analysis thanks to our cfDNA normal reference panel. The utility of

passenger mutations has been demonstrated in classifying tumor types based on tissue genomic data (<https://doi.org/10.1038/s41467-022-31666-w>, attached below). However, our study is the first attempt to leverage information embedded in passenger mutations for cancer diagnosis by cfDNA analysis.

[Important features for classifying tumor types based on tumor tissue WGS]

Regarding the epigenome model, most of previous studies focused on promoter regions and transcription factor binding sites. Here, we extended the scope to include distal regulatory regions in 25 different tissue types. It remained challenging to accurately identify nucleosome depleted regions (NDRs) by using conventional peak calling methods. As described in Supplementary Figure 2, we tested a different method and validated the biological relevance of this approach by using cell line data. In addition, the combinatorial projection of the density and length of read fragments onto the identified NDRs by means of the V-plot enabled us to make use of the convolutional neural network for three-dimensional image processing. We now have provided an in-detail description of these improvements made to our models in the introduction and discussion sections.

2. Understanding the mechanism of deep learning models is essential to trust the models and take advantage from the work for promoting related research. One possible approach is to use some interpretive methods for neural networks; another is to do some extended simulations. For example, what features in V-plot matter more in the diagnosis of cancer? Does all LMDs / NDRs contribute to the diagnosis, or important signals can be enriched into some specific gene sets? I suggest the authors to do more analysis to help the readers understand the significance of the algorithm.

We thank the reviewer for this constructive comment giving us an opportunity to provide a better interpretation of our models. A comprehensive characterization of model features was performed using attribution values derived by the integrative gradients method (<https://doi.org/10.48550/arXiv.1703.01365>). With regard to the genome model, we first show that somatic mutations from PCAWG tissue samples are enriched at regions with positive attribution scores compared to those with negative attribution scores for the cancer detection model (Figure 5A). We also show that tissue-specific high and low LMD regions tend to have positive and negative attribution scores, respectively (Figure 5B). In addition, high LMD regions of a particular cancer type had high attribution scores for the cfDNA samples of the matching cancer type, but not for the samples of other types (Figure 5C, upper). On the contrary, low LMD regions of a particular cancer type had low attribution scores only for the matched samples (Figure 5C, lower). These results indicate LMD regions derived from reference tumor tissue data play an important role in developing our prediction models for cancer detection and tissue-of-origin localization.

[Genome model interpretation results]

As for the epigenome model, we analyzed the attribution scores by means of the two-dimensional V-plot. In the cancer detection model, we found that fragment size distribution is important in distinguishing between normal and tumor samples (Figure 6A and B). In the cancer localization model, in contrast, fragment density across the NDRs (i.e., fragment distribution) plays an important role in classifying tumor site locations (Figure 6C). While depletion of cfDNA reads at the NDRs of a particular cancer type should increase the likelihood of assigning the given sample to the corresponding cancer type, enrichment of cfDNA reads at the NDRs of a particular cancer type is supposed to decrease the prediction probability for the given cancer type. As expected, negative attribution scores were assigned to reads mapping to the NDRs of a matching cancer type (Figure 6D-E).

[Epigenome model interpretation results]

3. The work used both MGI and Illumina platforms to sequence cfDNA. Can models trained on one platform be used to test data derived from another platform?

We thank the reviewer for this important comment. To perform external validation as suggested and comprehensively, we applied the combined model trained on one cohort to the training and validation dataset of the other cohorts. When the MGI-trained model was evaluated using the Illumina training cohort, Illumina validation cohort, and DELFI cohort, the performance was an ROC-AUC of 0.90, 0.88, and 0.83, respectively (Supplementary Figure 10C; attached below). When the Illumina-trained model was evaluated using the MGI training cohort, MGI validation cohort, and DELFI cohort, the performance was 0.84, 0.82, and 0.90, respectively (Supplementary Figure 10C; attached below). Additionally, the normal and tumor samples of the MGI cohort were clearly segregated by the prediction scores trained with the Illumina cohort, and *vice versa* (Supplementary Figure 10D; attached below).

[Performance of external validation]

To further confirm that there is no cohort/platform-specific feature bias, we performed hierarchical clustering of samples (columns) and features (rows) on the basis of feature attribution values. For both the MGI-trained and Illumina-trained model, we observe that the samples are separated not by the cohort (Illumina, MGI, or DELFI) but by the sample type (tumor or normal) (Figure 5D and Figure 6F). The same analysis was performed also by including the DELFI data, the results of which are attached below. The results including the DELFI data were not included in the manuscript because we were left with only two common tumor types by including the DELFI cohort.

[Clustering by feature attribution from cancer detection by the genome model (left) and epigenome model (right)]

[Clustering by feature attribution from cancer localization by the genome model (left) and epigenome model (right)]

4. The cost of sequencing is an important issue of cfDNA cancer diagnosis. The data derived from the MGI platform are around 5X, which seems to be a costly depth. I suggest the authors to do some downsampling to investigate the relationship between sequencing depth and precision.

To address this interesting point, we examined how model performance can be affected by downsampling. For the MGI cohort data, we carried out 3x and 1x downsampling. Although higher sequencing depth tended to guarantee higher model performance, 3x downsampling did not considerably undermine the performance of the model developed with the original data. Thanks to this helpful comment, we have learned that 3x sequencing depth can be a practical option.

[Downsampling performance]

5. The usage of K562 and GM12878 cell line data seems to be unclear. What's the relationship between the data and the downstream analysis?

As described in our response to comment #1, one of the advantages of our epigenome

model is that it uses biologically meaningful NDRs identified across the whole genome beyond gene promoters. Using the GM12878 and K562 cell lines, we tested the performance of the ATAC-seq peak calling method of HMMRATAC by examining cell type specificity and interrogating various histone modification data. All detailed analysis NDR results including these are provided in Supplementary Figure 2 as attached. Based on these pilot test results, we applied our NDR calling pipeline to 431 tissue ATAC-seq data resulting in 25 tissue-specific NDRs selected by EdgeR.

[Pilot tests for NDR identification using cell line data]

6. The authors used five-fold cross-validation to train and test the model. Though the approach seems to be reasonable, I am still curious about why the authors don't used a more straightforward approach to train, validate and test the model with such a large dataset. The authors seem to train a lot parallel models in the whole approach, but which model was used to test the external DELFI dataset? Does the choice of hyper-parameters matter a lot to the final results? If I were a user of the algorithm, how should I merge these parallel models? I suggest the authors to explain the training and testing approach more systematically to help the readers understand the work better and guarantee that there is no information leakage in the approach.

We agree with the reviewer that our dataset is large enough for a simple split. However, there are two reasons why we use 5-fold cross-validation. The first reason is to avoid the curse of dimensionality. Data of more than 1,000 samples is large enough in the biological field, but the number of features we use to train our model is much higher. We proceeded with 5-fold cross-validation to obtain the robustness of our models while avoiding the possibility of overfitting that can occur when the number of features is greater than the number of training samples. The second reason is to directly compare our models with the DELFI algorithm on their dataset. In the DELFI paper (<https://doi.org/10.1038/s41586-019-1272-6>), they trained the model using cross-validation. In order to make a fair comparison using the prediction probabilities of the samples provided in the DELFI paper, we followed the DELFI model training procedure as much as possible (Supplementary Figure 5).

[DELFI model training procedure]

We apologize for missing the details of our model training and validation processes. As mentioned above, we conducted model training in a similar way to the DELFI paper. Briefly, in the training step, we separated our samples into 5-fold cross-validation (CV) with stratification. Each CV underwent 200 hyperparameter searches to determine the best hyperparameter, and the model was trained by repeating 30 times using the selected best hyperparameters, and among them, the model with the lowest validation loss was selected as the final model. For each CV, the test set was predicted using the final model of that CV, and the performance was evaluated using the test prediction score of all samples. In the validation step, we evaluated the performance

by calculating the average of prediction scores obtained from the 5 models. The model training and validation processes are now described in detail in the text and Supplementary Figure 3B-C.

Model training process

Model prediction process

[Our model training and validation processes]

Reviewer #4

Reviewer #4 – General comments:

In this study, the authors performed cell-free DNA (cfDNA) whole genome sequencing to generate two test datasets including 2543 patient samples from nine different cancer types and 1241 normal control samples, and a reference dataset for background variant filtering based on 20529 samples from low-depth healthy subjects. An external cfDNA data set consisting of 208 cancer samples and 214 normal controls was also used for additional evaluation. The algorithm incorporates as model criteria the distribution of mutations in tumor tissue and cancer type-specific profiles of chromatin tissue, achieving very high accuracy in cancer detection and tissue origin localization. This integrated model was able to detect early-stage cancers, including pancreatic cancer, with high sensitivity comparable to late-stage cancers. In addition, interpretation of the model revealed the contribution of genomic and epigenomic features to different types of cancer. This methodology could lay the groundwork for accurate cfDNA-based cancer diagnosis, especially at early stages. The authors are working on an important research topic and have obtained some interesting results. On the other hand, there are several problems, and I recommend a major revision of the manuscripts, paying attention to the following points.

We thank the reviewer for acknowledging the importance of the subject matter and the implications of this work. We have made efforts to address the points raised by the reviewer, especially by providing clinical information, and now feel that our paper has been improved considerably thanks to the constructive comments.

Reviewer #4 – Major comments:

1. Despite the complexity of the research, the number of main figures is small and the text in each figure is small, making it difficult to grasp the overall picture of the research. It would be better to increase the number of main figures and make the letters in the figures larger.

We apologize for the lack of clarity. According to the reviewer's comment, we now have added main and supplementary figures, in particular regarding the illustration of model concepts and the results of model interpretation. We have also increased the font size of the figures for better readability.

2. Deep neural networks were used in this study, but it is difficult to understand the details in the current version. The details should be shown in Fig. 1 or elsewhere, including the structure of the algorithm.

We appreciate this constructive comment. We have modified Figure 1 to better describe the concept and overall process of the genome and epigenome models, and also generated Supplementary Figure 3 to describe the model training process, structure, and hyperparameters in detail. Briefly, the genome model consisted of

features related to local mutation density and variant type for passenger mutations and conducted training with deep neural networks. The epigenome model consisted of three-dimensional V-plot images representing fragment density and fragment size across 25 tissue-specific NDRs and conducted training with convolutional neural networks.

[Schematic diagram of the genome, epigenome, and combined model]

[Model structure and hyperparameter list]

3. The most serious problem with this article is that, despite the fact that it is a medical and clinical-oriented study, it provides no information about the research institution or the institution from which the clinical specimens were taken. The authors are not even sure if they are cancer experts to begin with, although they have done some clinical-leaning research on cancer in particular. Given the above, it is also impossible to determine whether the results of the study have been properly interpreted. To be honest, the limited information on the research group in this paper makes it difficult to evaluate the article's content. When publishing such clinically oriented papers, it should be possible to determine the type of professional (clinician, medical researcher, informatics researcher) from the peer review stage.

4. Related to 3, the current manuscript is unclear in which region (country) the study was conducted. From the MATERIALS AND METHODS, I have determined that it is probably a Chinese research group, but in that case, the genetic background by ethnic group should also be noted. A number of studies have reported results indicating that genetic backgrounds in Western countries differ significantly from those in Asian regions to begin with. A method of analysis that mixes up genetic backgrounds, as in this case, may lead to misinterpretation of the results.

We apologize for missing the critical information. On the basis of blind review, we masked the author information during our initial submission. For clarification, this research was conducted in South Korea. To demonstrate the credibility of our research, we now have disclosed author information and the identities of institutions from which the clinical specimens were taken in the method section (page 25, line 4). For multi-cancer detection, patient samples of 9 cancer types were recruited from 6 institutions of South Korea, and their clinical information was obtained and processed by the professional clinicians of these institutions. The samples were sent in streck tubes to GC Genome Corporation (<https://www.gc-genome.com/>), where whole-genome sequencing of cfDNA was performed using the MGI and Illumina platform. The sequencing data was analyzed at a computational biology laboratory of KAIST (<http://omics.kaist.ac.kr>). We provide detail clinical information in Supplementary Table 7.

We understand the reviewer's concern regarding genetic background. To minimize potential bias, we leverage our own germline variant filtering data based on 20,529 low-depth healthy samples of the same ethnicity (processed at GC Genome). We believe that unlike germline variants, somatic mutations are not heavily dependent on ethnicity. According to the results of external validation, applying our models trained on the Korean samples to the DEFLI data does not particularly undermine prediction performance (see below).

[Application of the MGI/Illumina models to the DELFI cohort]

While putting together clinical information in Supplementary Table 7, it was brought to our attention that the Illumina breast cancer samples, unlike other fresh data directly provided by the hospitals, were commercially purchased. Our inspection of these samples revealed that the distribution of fragment size from these samples was quite different from that from the other samples. Therefore, we decided to exclude these potentially problematic samples from further analyses. Thanks to the reviewer's comment, we were able to conduct further quality control.

5. To be honest, my first impression from reading the article is that the AUC values in Figure 2 are too high, which may have caused over fitting. It is difficult to determine accuracy from retrospective studies alone, and it would be impossible to properly determine usefulness without conducting prospective studies.

We agree with the reviewer that prospective studies can be useful in verifying model performance. Unfortunately, a prospective study requires enormous amount of planning, clinical arrangement, and resources and is way beyond the scope of this work. Instead, we put a lot of effort into proving that our model performance is not merely the result of overfitting.

First, in the process of sample collection, we set aside samples acquired before a certain time point as the training set and assigned the subsequent samples as the validation set, practically mimicking a prospective approach. Prediction performance evaluated on the validation set indicates no sign of overfitting to the training data (Figure 2A-D).

Second, to demonstrate that our model is not overfitted but trained with biological relevance, we conducted model interpretation using the integrated gradients method (<https://doi.org/10.48550/arXiv.1703.01365>). With regard to the genome model, we

first show that somatic mutations from PCAWG tissue samples are enriched at regions with positive attribution scores compared to those with negative attribution scores for the cancer detection model (Figure 5A). We also show that tissue-specific high and low LMD regions tend to have positive and negative attribution scores, respectively (Figure 5B). In addition, high LMD regions of a particular cancer type had high attribution scores for the cfDNA samples of the matching cancer type, but not for the samples of other types (Figure 5C, upper). On the contrary, low LMD regions of a particular cancer type had low attribution scores specifically for the matched samples (Figure 5C, lower). These results indicate LMD regions derived from reference tumor tissue data play an important role in developing our prediction models for cancer detection and tissue-of-origin localization.

[Genome model interpretation results]

As for the epigenome model, we analyzed the attribution scores by means of the two-dimensional V-plot. In the cancer detection model, we found that fragment size distribution is important in distinguishing between normal and tumor samples (Figure 6A and B). In the cancer localization model, in contrast, fragment density across the NDRs (i.e., fragment distribution) plays an important role in classifying tumor site locations (Figure 6C). While depletion of cfDNA reads at the NDRs of a particular cancer type should increase the likelihood of assigning the given sample to the corresponding cancer type, enrichment of cfDNA reads at the NDRs of a particular cancer type is supposed to decrease the prediction probability for the given cancer type. As expected, negative attribution scores were assigned to reads mapping to the NDRs of a matching cancer type (Figure 6D-E).

[Epigenome model interpretation results]

Finally, we checked the correlation of the feature importance values measured with the MGI, Illumina, and DELFI cohorts because overfitting would have resulted in specific patterns of feature importance tailored to each training cohort. Thus, we performed hierarchical clustering of samples (columns) and features (rows) on the basis of feature attribution values. For both the MGI-trained and Illumina-trained model, we observe that the samples are separated not by the cohort (Illumina, MGI, or DELFI) but by the sample type (tumor or normal) (Figure 5D and Figure 6F). The same analysis was performed also by including the DELFI data, the results of which are attached below. The results including the DELFI data were not included in the manuscript because we were left with only two common tumor types by including the DELFI cohort.

[Clustering by feature attribution from cancer detection by the genome model (left) and epigenome model (right)]

[Clustering by feature attribution from cancer localization by the genome model (left) and epigenome model (right)]

6. In the judgment of cancer experts, the molecular mechanisms of cancer development differ greatly in different organs. It is difficult to determine whether the method of training all nine types of cancer in a jumble as in this case is really useful. In particular, in Figures 2E and F, the colon, biliary tract, and head and neck data are almost statistically meaningless because the number of samples is small and the error bars are too large. I do not understand the significance of presenting such data. Similarly, the data in NA in Figure 2E have too few samples and very large error bars. I find the data to be completely meaningless from a scientific standpoint.

We agree with the reviewer that each type of cancer has different characteristics. However, multi-cancer early detection is a primary goal of cancer screening (New

genomic technologies for multi-cancer early detection: Rethinking the scope of cancer screening. *Cancer Cell* 40:109–113), and previous models as well as ours use common features that appear regardless of cancer types such as variant and fragment properties. As our cancer detection models were developed on the basis of such pan-cancer features albeit centered on the nine cancer types, we believe that the prediction results for cancer types with a small number of samples (e.g., colon, biliary tract, and head and neck) still hold biological meaning. In contrast, predicting tumor origin localization may be problematic with these cancer types having a small number of cases in terms of statistical confidence as the reviewer pointed out. Therefore, we excluded cancer types with fewer than 40 samples from our tissue-of-origin prediction.

REVIEWERS' COMMENTS

Reviewer #1 (Remarks to the Author):

All concerns are addressed.

Reviewer #3 (Remarks to the Author):

The authors have well addressed all my concerns. I have no more questions about this work. Besides, I found a typo in Figure 1: "gremlin".

Reviewer #4 (Remarks to the Author):

Basically, I feel that the authors have effectively addressed the criticisms raised in the initial review. I recommend publication.

Reviewer #1

Reviewer #1 – Major comments:

1. Not sure it should be called integrative modeling, since it is only calculating the average score from independent genome model and epigenome model.

Our apologies for the confusion. By integrative modeling, we referred to our approach that combines various public data and our own datasets (normal cfDNA reference panel consisting of 20,529 samples, PCAWG, TCGA, ENCODE, GEO, dbSNP, 1000 Genomes, HapMap, ExAC, and gnomAD). Especially, modeling of the tumor tissue mutation and chromatin profiles from PCAWG and TCGA played a critical role in developing our prediction models. Our normal reference panel also played an essential role in filtering artifactual variants in developing the genome model. For clarification, we have modified the Figure 1 as follows.

[Pipeline of integrative modeling with reference datasets]

2. How the DELFI used samples were processed is not mentioned anywhere in the manuscript.

We apologize for the lack of detailed information. We have now added the description of DELFI cohort processing procedures in the method section as follows (page 34, line 11):

Processing of the DELFI cohort data

We used the DELFI dataset with 1-2x cfDNA WGS of 214 healthy samples and 208 cancer patients to validate our algorithm. Cancer patient samples include breast (n=54), pancreatic (n=34), ovarian (n=28), colorectal (n=27), gastric (n=27), lung (n=12), and bile duct cancer (n=26). Following the approval of their Data Access Committee (DAC), duplicate marked bam files of the DELFI dataset were obtained from European Genome-Phenome Archive (EGA). Genome, epigenome, cnv, fragpattern, and fragsize input features were processed using duplicate marked bam as described in the sections "Genome model input processing and training", "Epigenome model input processing and training" and "Cnv, fragpattern, and fragsize model input processing and training".

3. The high sensitivity and specificity are impressive. But here are some model evaluation-related concerns: In last paragraph of Page.20, author mentioned that the best performance from 30 times running was selected as the validation performance, but it will be important for the readers to know the variance of the 30 models' performance to determine the robustness of the model.

We thank the reviewer for this constructive comment giving us an opportunity to prove the robustness of our models. To address this comment, we examined the variance of the 30 models' performance in terms of the loss and AUC for the genome and epigenome model in cancer detection and tissue-of-origin localization as follows. These results are provided in Supplementary Figures 4 and 5 (for cancer detection and localization, respectively). Although the initial seed was changed during the 30 repetitions, no significant difference in performance was observed.

[Performance variation in cancer detection]

A: Genome model on MGI training cohort

B: Epigenome model on MGI training cohort

C: Genome model on Illumina training cohort

D: Epigenome model on Illumina training cohort

[Performance variation in tissue-of-origin localization]

A: Genome model on MGI training cohort

B: Epigenome model on MGI training cohort

C: Genome model on Illumina training cohort

D: Epigenome model on Illumina training cohort

4. It is not clear how results shown in Supplementary figure 4 was derived. How the evaluation was done on DELFI dataset is not mentioned in methods part. e.g., with so high a sensitivity and specificity, is it cross-validation results (meaning model trained on DELFI samples again) or just test results predicted using MGI/Illumina-trained model? If the latter, which platform-trained models were used to predict DELFI samples? It is critical results determining the model performance, and should be described to help readers understanding the whole evaluation process.

Our apologies for the confusion. In the original Supplementary Figure 4 (currently Supplementary Figure 6), the DELFI cohort data was used for model training. By going through the same training and cross-validation processes, we were able to fairly compare the performance of our models with the DELFI algorithm itself. The details of these processes involving the DELFI data are provided in Supplementary Figure 3A of the revised manuscript as follows (page 34, line 24):

Model training using the training cohort

Each training cohort (MGI, Illumina, and DELFI cohort) for the MGI and Illumina sequencing

platforms was partitioned into five groups for the application of the stratified five-fold cross-validation. At each iteration, four groups in the training set were further divided into three training sets and one validation set. Using this method, all samples were given a test prediction score from each model. Using the test prediction score of all training cohort samples, we calculated the ROC-AUC score for cancer detection and the accuracy score for tissue-of-origin localization. The confidence interval for sensitivity was calculated from 1,000 bootstrap samplings with replicates at 95%, 98%, and 99% specificity. The cancer localization model was developed by using either all cancer samples in the training cohorts or the cancer samples correctly identified by the combined cancer detection model with 98% specificity. The number of correctly predicted samples was 1,188 out of 1,359 for the MGI cohort and 644 out of 940 samples for the Illumina cohort.

During the revision, we also used the DELFI data as the test dataset for our MGI/Illumina-trained models. The results are described in our response to comment #5 below.

5. It would be a more comprehensive evaluation if the MGI-trained model can be applied in Illumina samples.

We appreciate this critical comment. To address this point comprehensively, we performed external validation by applying the combined model trained on one cohort to the training and validation dataset of the other cohorts. When the MGI-trained model was evaluated using the Illumina training cohort, Illumina validation cohort, and DELFI cohort, the performance was an ROC-AUC of 0.90, 0.88, and 0.83, respectively (Supplementary Figure 10C; attached below). When the Illumina-trained model was evaluated using the MGI training cohort, MGI validation cohort, and DELFI cohort, the performance was 0.84, 0.82, and 0.90, respectively (Supplementary Figure 10C; attached below). Additionally, the normal and tumor samples of the MGI cohort were clearly segregated by the prediction scores trained with the Illumina cohort, and *vice versa* (Supplementary Figure 10D; attached below).

[Performance of external validation]

6. In model interpretation, the consistence between MGI-trained and Illumina-trained model feature importance could to be evaluated. This is helpful in determining whether models are overfitting their own datasets, or discovered real early cancer genome and epigenome changes.

We appreciate this constructive comment. To address this point systematically, we performed hierarchical clustering of samples (columns) and features (rows) on the basis of feature attribution values. For both the MGI-trained and Illumina-trained model, we observe that the samples are separated not by the cohort (Illumina, MGI, or DELFI) but by the sample type (tumor or normal) (Figure 5D and Figure 6F). The same analysis was performed also by including the DELFI data, the results of which are attached below. The results including the DELFI data were not included in the manuscript because we were left with only two common tumor types by including the DELFI cohort.

[Clustering by feature attribution from cancer detection by the genome model (left) and epigenome model (right)]

[Clustering by feature attribution from cancer localization by the genome model (left) and epigenome model (right)]

Reviewer #3

Reviewer #3 – General comments:

In this paper, the authors proposed a work that tried to diagnose cancer through cfDNA WGS data by integrating both genome and epigenome features. The data size and performance of this work is impressive, but there are some key issues that the authors didn't address well. I have some concerns about the significance and methods of this work.

We thank the reviewer for the detailed comments and for acknowledging the importance of the subject matter. We have made efforts to address the points raised by the reviewer, especially by describing model interpretation more comprehensively, and now feel that our paper has been improved considerably thanks to the constructive comments.

Reviewer #3 – Major comments:

1. The key significance of the work seems to be the algorithm that integrate both genome and epigenome features. However, there are few details about the algorithm. For example, no figure about the algorithm was shown in the article, which put obstacles to the readers to understand the algorithm. A pipeline should be provided to assess the whole algorithm process, e.g. the detail of CNN and the usage of different data. Actually, it is intuitional to merge mutation and fragment information of cfDNA to enhance the accuracy of diagnosis, and there are many works to deal with the issues. How and why the authors develop their algorithm? What's the difference between their algorithm and other works that deal with genomes, epigenomes as well as integration of them? What can other researchers learn from this work to deal with their own problems? I think the authors should explain these issues carefully in both introduction and discussion, which could benefit more to the scientific community.

We apologize for the lack of detailed information. We now have provided further clarifications on model building procedures, model structures, and learning hyperparameters in the form of schematic illustrations (Figure 1 and Supplementary Figure 3A; attached below).

[Schematic of model building procedures (Figure 1)]

[Schematic of training processes and structures (Supplementary Figure 3A)]

Although some previous studies also used mutation and fragmentation features, our study significantly improved the performance of cancer diagnosis by modeling of large-scale tumor tissue data. As for the genome model using variant features, most previous studies focused on driver mutations of oncogenes and tumor suppressor genes such as TP53 and KRAS. In contrast, we were able to also include passenger mutations in our analysis thanks to our cfDNA normal reference panel. The utility of

passenger mutations has been demonstrated in classifying tumor types based on tissue genomic data (<https://doi.org/10.1038/s41467-022-31666-w>, attached below). However, our study is the first attempt to leverage information embedded in passenger mutations for cancer diagnosis by cfDNA analysis.

[Important features for classifying tumor types based on tumor tissue WGS]

Regarding the epigenome model, most of previous studies focused on promoter regions and transcription factor binding sites. Here, we extended the scope to include distal regulatory regions in 25 different tissue types. It remained challenging to accurately identify nucleosome depleted regions (NDRs) by using conventional peak calling methods. As described in Supplementary Figure 2, we tested a different method and validated the biological relevance of this approach by using cell line data. In addition, the combinatorial projection of the density and length of read fragments onto the identified NDRs by means of the V-plot enabled us to make use of the convolutional neural network for three-dimensional image processing. We now have provided an in-detail description of these improvements made to our models in the introduction and discussion sections.

2. Understanding the mechanism of deep learning models is essential to trust the models and take advantage from the work for promoting related research. One possible approach is to use some interpretive methods for neural networks; another is to do some extended simulations. For example, what features in V-plot matter more in the diagnosis of cancer? Does all LMDs / NDRs contribute to the diagnosis, or important signals can be enriched into some specific gene sets? I suggest the authors to do more analysis to help the readers understand the significance of the algorithm.

We thank the reviewer for this constructive comment giving us an opportunity to provide a better interpretation of our models. A comprehensive characterization of model features was performed using attribution values derived by the integrative gradients method (<https://doi.org/10.48550/arXiv.1703.01365>). With regard to the genome model, we first show that somatic mutations from PCAWG tissue samples are enriched at regions with positive attribution scores compared to those with negative attribution scores for the cancer detection model (Figure 5A). We also show that tissue-specific high and low LMD regions tend to have positive and negative attribution scores, respectively (Figure 5B). In addition, high LMD regions of a particular cancer type had high attribution scores for the cfDNA samples of the matching cancer type, but not for the samples of other types (Figure 5C, upper). On the contrary, low LMD regions of a particular cancer type had low attribution scores only for the matched samples (Figure 5C, lower). These results indicate LMD regions derived from reference tumor tissue data play an important role in developing our prediction models for cancer detection and tissue-of-origin localization.

[Genome model interpretation results]

As for the epigenome model, we analyzed the attribution scores by means of the two-dimensional V-plot. In the cancer detection model, we found that fragment size distribution is important in distinguishing between normal and tumor samples (Figure 6A and B). In the cancer localization model, in contrast, fragment density across the NDRs (i.e., fragment distribution) plays an important role in classifying tumor site locations (Figure 6C). While depletion of cfDNA reads at the NDRs of a particular cancer type should increase the likelihood of assigning the given sample to the corresponding cancer type, enrichment of cfDNA reads at the NDRs of a particular cancer type is supposed to decrease the prediction probability for the given cancer type. As expected, negative attribution scores were assigned to reads mapping to the NDRs of a matching cancer type (Figure 6D-E).

[Epigenome model interpretation results]

3. The work used both MGI and Illumina platforms to sequence cfDNA. Can models trained on one platform be used to test data derived from another platform?

We thank the reviewer for this important comment. To perform external validation as suggested and comprehensively, we applied the combined model trained on one cohort to the training and validation dataset of the other cohorts. When the MGI-trained model was evaluated using the Illumina training cohort, Illumina validation cohort, and DELFI cohort, the performance was an ROC-AUC of 0.90, 0.88, and 0.83, respectively (Supplementary Figure 10C; attached below). When the Illumina-trained model was evaluated using the MGI training cohort, MGI validation cohort, and DELFI cohort, the performance was 0.84, 0.82, and 0.90, respectively (Supplementary Figure 10C; attached below). Additionally, the normal and tumor samples of the MGI cohort were clearly segregated by the prediction scores trained with the Illumina cohort, and *vice versa* (Supplementary Figure 10D; attached below).

[Performance of external validation]

To further confirm that there is no cohort/platform-specific feature bias, we performed hierarchical clustering of samples (columns) and features (rows) on the basis of feature attribution values. For both the MGI-trained and Illumina-trained model, we observe that the samples are separated not by the cohort (Illumina, MGI, or DELFI) but by the sample type (tumor or normal) (Figure 5D and Figure 6F). The same analysis was performed also by including the DELFI data, the results of which are attached below. The results including the DELFI data were not included in the manuscript because we were left with only two common tumor types by including the DELFI cohort.

[Clustering by feature attribution from cancer detection by the genome model (left) and epigenome model (right)]

[Clustering by feature attribution from cancer localization by the genome model (left) and epigenome model (right)]

4. The cost of sequencing is an important issue of cfDNA cancer diagnosis. The data derived from the MGI platform are around 5X, which seems to be a costly depth. I suggest the authors to do some downsampling to investigate the relationship between sequencing depth and precision.

To address this interesting point, we examined how model performance can be affected by downsampling. For the MGI cohort data, we carried out 3x and 1x downsampling. Although higher sequencing depth tended to guarantee higher model performance, 3x downsampling did not considerably undermine the performance of the model developed with the original data. Thanks to this helpful comment, we have learned that 3x sequencing depth can be a practical option.

[Downsampling performance]

5. The usage of K562 and GM12878 cell line data seems to be unclear. What's the relationship between the data and the downstream analysis?

As described in our response to comment #1, one of the advantages of our epigenome

model is that it uses biologically meaningful NDRs identified across the whole genome beyond gene promoters. Using the GM12878 and K562 cell lines, we tested the performance of the ATAC-seq peak calling method of HMMRATAC by examining cell type specificity and interrogating various histone modification data. All detailed analysis results including these are provided in Supplementary Figure 2 as attached. Based on these pilot test results, we applied our NDR calling pipeline to 431 tissue ATAC-seq data resulting in 25 tissue-specific NDRs selected by EdgeR.

[Pilot tests for NDR identification using cell line data]

6. The authors used five-fold cross-validation to train and test the model. Though the approach seems to be reasonable, I am still curious about why the authors don't used a more straightforward approach to train, validate and test the model with such a large dataset. The authors seem to train a lot parallel models in the whole approach, but which model was used to test the external DELFI dataset? Does the choice of hyper-parameters matter a lot to the final results? If I were a user of the algorithm, how should I merge these parallel models? I suggest the authors to explain the training and testing approach more systematically to help the readers understand the work better and guarantee that there is no information leakage in the approach.

We agree with the reviewer that our dataset is large enough for a simple split. However, there are two reasons why we use 5-fold cross-validation. The first reason is to avoid the curse of dimensionality. Data of more than 1,000 samples is large enough in the biological field, but the number of features we use to train our model is much higher. We proceeded with 5-fold cross-validation to obtain the robustness of our models while avoiding the possibility of overfitting that can occur when the number of features is greater than the number of training samples. The second reason is to directly compare our models with the DELFI algorithm on their dataset. In the DELFI paper (<https://doi.org/10.1038/s41586-019-1272-6>), they trained the model using cross-validation. In order to make a fair comparison using the prediction probabilities of the samples provided in the DELFI paper, we followed the DELFI model training procedure as much as possible (Supplementary Figure 5).

[DELFI model training procedure]

We apologize for missing the details of our model training and validation processes. As mentioned above, we conducted model training in a similar way to the DELFI paper. Briefly, in the training step, we separated our samples into 5-fold cross-validation (CV) with stratification. Each CV underwent 200 hyperparameter searches to determine the best hyperparameter, and the model was trained by repeating 30 times using the selected best hyperparameters, and among them, the model with the lowest validation loss was selected as the final model. For each CV, the test set was predicted using the final model of that CV, and the performance was evaluated using the test prediction score of all samples. In the validation step, we evaluated the performance

by calculating the average of prediction scores obtained from the 5 models. The model training and validation processes are now described in detail in the text and Supplementary Figure 3B-C.

Model training process

Model prediction process

[Our model training and validation processes]

Reviewer #4

Reviewer #4 – General comments:

In this study, the authors performed cell-free DNA (cfDNA) whole genome sequencing to generate two test datasets including 2543 patient samples from nine different cancer types and 1241 normal control samples, and a reference dataset for background variant filtering based on 20529 samples from low-depth healthy subjects. An external cfDNA data set consisting of 208 cancer samples and 214 normal controls was also used for additional evaluation. The algorithm incorporates as model criteria the distribution of mutations in tumor tissue and cancer type-specific profiles of chromatin tissue, achieving very high accuracy in cancer detection and tissue origin localization. This integrated model was able to detect early-stage cancers, including pancreatic cancer, with high sensitivity comparable to late-stage cancers. In addition, interpretation of the model revealed the contribution of genomic and epigenomic features to different types of cancer. This methodology could lay the groundwork for accurate cfDNA-based cancer diagnosis, especially at early stages. The authors are working on an important research topic and have obtained some interesting results. On the other hand, there are several problems, and I recommend a major revision of the manuscripts, paying attention to the following points.

We thank the reviewer for acknowledging the importance of the subject matter and the implications of this work. We have made efforts to address the points raised by the reviewer, especially by providing clinical information, and now feel that our paper has been improved considerably thanks to the constructive comments.

Reviewer #4 – Major comments:

1. Despite the complexity of the research, the number of main figures is small and the text in each figure is small, making it difficult to grasp the overall picture of the research. It would be better to increase the number of main figures and make the letters in the figures larger.

We apologize for the lack of clarity. According to the reviewer's comment, we now have added main and supplementary figures, in particular regarding the illustration of model concepts and the results of model interpretation. We have also increased the font size of the figures for better readability.

2. Deep neural networks were used in this study, but it is difficult to understand the details in the current version. The details should be shown in Fig. 1 or elsewhere, including the structure of the algorithm.

We appreciate this constructive comment. We have modified Figure 1 to better describe the concept and overall process of the genome and epigenome models, and also generated Supplementary Figure 3 to describe the model training process, structure, and hyperparameters in detail. Briefly, the genome model consisted of

features related to local mutation density and variant type for passenger mutations and conducted training with deep neural networks. The epigenome model consisted of three-dimensional V-plot images representing fragment density and fragment size across 25 tissue-specific NDRs and conducted training with convolutional neural networks.

[Schematic diagram of the genome, epigenome, and combined model]

[Model structure and hyperparameter list]

3. The most serious problem with this article is that, despite the fact that it is a medical and clinical-oriented study, it provides no information about the research institution or the institution from which the clinical specimens were taken. The authors are not even sure if they are cancer experts to begin with, although they have done some clinical-leaning research on cancer in particular. Given the above, it is also impossible to determine whether the results of the study have been properly interpreted. To be honest, the limited information on the research group in this paper makes it difficult to evaluate the article's content. When publishing such clinically oriented papers, it should be possible to determine the type of professional (clinician, medical researcher, informatics researcher) from the peer review stage.

4. Related to 3, the current manuscript is unclear in which region (country) the study was conducted. From the MATERIALS AND METHODS, I have determined that it is probably a Chinese research group, but in that case, the genetic background by ethnic group should also be noted. A number of studies have reported results indicating that genetic backgrounds in Western countries differ significantly from those in Asian regions to begin with. A method of analysis that mixes up genetic backgrounds, as in this case, may lead to misinterpretation of the results.

We apologize for missing the critical information. On the basis of blind review, we masked the author information during our initial submission. For clarification, this research was conducted in South Korea. To demonstrate the credibility of our research, we now have disclosed author information and the identities of institutions from which the clinical specimens were taken in the method section (page 25, line 4). For multi-cancer detection, patient samples of 9 cancer types were recruited from 6 institutions of South Korea, and their clinical information was obtained and processed by the professional clinicians of these institutions. The samples were sent in streck tubes to GC Genome Corporation (<https://www.gc-genome.com/>), where whole-genome sequencing of cfDNA was performed using the MGI and Illumina platform. The sequencing data was analyzed at a computational biology laboratory of KAIST (<http://omics.kaist.ac.kr>). We provide detail clinical information in Supplementary Table 7.

We understand the reviewer's concern regarding genetic background. To minimize potential bias, we leverage our own germline variant filtering data based on 20,529 low-depth healthy samples of the same ethnicity (processed at GC Genome). We believe that unlike germline variants, somatic mutations are not heavily dependent on ethnicity. According to the results of external validation, applying our models trained on the Korean samples to the DEFLI data does not particularly undermine prediction performance (see below).

[Application of the MGI/Illumina models to the DELFI cohort]

While putting together clinical information in Supplementary Table 7, it was brought to our attention that the Illumina breast cancer samples, unlike other fresh data directly provided by the hospitals, were commercially purchased. Our inspection of these samples revealed that the distribution of fragment size from these samples was quite different from that from the other samples. Therefore, we decided to exclude these potentially problematic samples from further analyses. Thanks to the reviewer's comment, we were able to conduct further quality control.

5. To be honest, my first impression from reading the article is that the AUC values in Figure 2 are too high, which may have caused over fitting. It is difficult to determine accuracy from retrospective studies alone, and it would be impossible to properly determine usefulness without conducting prospective studies.

We agree with the reviewer that prospective studies can be useful in verifying model performance. Unfortunately, a prospective study requires enormous amount of planning, clinical arrangement, and resources and is way beyond the scope of this work. Instead, we put a lot of effort into proving that our model performance is not merely the result of overfitting.

First, in the process of sample collection, we set aside samples acquired before a certain time point as the training set and assigned the subsequent samples as the validation set, practically mimicking a prospective approach. Prediction performance evaluated on the validation set indicates no sign of overfitting to the training data (Figure 2A-D).

Second, to demonstrate that our model is not overfitted but trained with biological relevance, we conducted model interpretation using the integrated gradients method (<https://doi.org/10.48550/arXiv.1703.01365>). With regard to the genome model, we

first show that somatic mutations from PCAWG tissue samples are enriched at regions with positive attribution scores compared to those with negative attribution scores for the cancer detection model (Figure 5A). We also show that tissue-specific high and low LMD regions tend to have positive and negative attribution scores, respectively (Figure 5B). In addition, high LMD regions of a particular cancer type had high attribution scores for the cfDNA samples of the matching cancer type, but not for the samples of other types (Figure 5C, upper). On the contrary, low LMD regions of a particular cancer type had low attribution scores specifically for the matched samples (Figure 5C, lower). These results indicate LMD regions derived from reference tumor tissue data play an important role in developing our prediction models for cancer detection and tissue-of-origin localization.

[Genome model interpretation results]

As for the epigenome model, we analyzed the attribution scores by means of the two-dimensional V-plot. In the cancer detection model, we found that fragment size distribution is important in distinguishing between normal and tumor samples (Figure 6A and B). In the cancer localization model, in contrast, fragment density across the NDRs (i.e., fragment distribution) plays an important role in classifying tumor site locations (Figure 6C). While depletion of cfDNA reads at the NDRs of a particular cancer type should increase the likelihood of assigning the given sample to the corresponding cancer type, enrichment of cfDNA reads at the NDRs of a particular cancer type is supposed to decrease the prediction probability for the given cancer type. As expected, negative attribution scores were assigned to reads mapping to the NDRs of a matching cancer type (Figure 6D-E).

[Epigenome model interpretation results]

Finally, we checked the correlation of the feature importance values measured with the MGI, Illumina, and DELFI cohorts because overfitting would have resulted in specific patterns of feature importance tailored to each training cohort. Thus, we performed hierarchical clustering of samples (columns) and features (rows) on the basis of feature attribution values. For both the MGI-trained and Illumina-trained model, we observe that the samples are separated not by the cohort (Illumina, MGI, or DELFI) but by the sample type (tumor or normal) (Figure 5D and Figure 6F). The same analysis was performed also by including the DELFI data, the results of which are attached below. The results including the DELFI data were not included in the manuscript because we were left with only two common tumor types by including the DELFI cohort.

[Clustering by feature attribution from cancer detection by the genome model (left) and epigenome model (right)]

[Clustering by feature attribution from cancer localization by the genome model (left) and epigenome model (right)]

6. In the judgment of cancer experts, the molecular mechanisms of cancer development differ greatly in different organs. It is difficult to determine whether the method of training all nine types of cancer in a jumble as in this case is really useful. In particular, in Figures 2E and F, the colon, biliary tract, and head and neck data are almost statistically meaningless because the number of samples is small and the error bars are too large. I do not understand the significance of presenting such data. Similarly, the data in NA in Figure 2E have too few samples and very large error bars. I find the data to be completely meaningless from a scientific standpoint.

We agree with the reviewer that each type of cancer has different characteristics. However, multi-cancer early detection is a primary goal of cancer screening (New

genomic technologies for multi-cancer early detection: Rethinking the scope of cancer screening. *Cancer Cell* 40:109–113), and previous models as well as ours use common features that appear regardless of cancer types such as variant and fragment properties. As our cancer detection models were developed on the basis of such pan-cancer features albeit centered on the nine cancer types, we believe that the prediction results for cancer types with a small number of samples (e.g., colon, biliary tract, and head and neck) still hold biological meaning. In contrast, predicting tumor origin localization may be problematic with these cancer types having a small number of cases in terms of statistical confidence as the reviewer pointed out. Therefore, we excluded cancer types with fewer than 40 samples from our tissue-of-origin prediction.

Reviewer #1

All concerns are addressed.

We thank the reviewer for the positive feedback.

Reviewer #2

The authors have well addressed all my concerns. I have no more questions about this work. Besides, I found a typo in Figure 1: “gremlin”.

We appreciate the reviewer for the positive feedback. Additionally, we edited a type in Figure 1.

Reviewer #4

Basically, I fell that the authors have effectively addressed the criticisms raised in the initial review. I recommend publication.

We appreciate the reviewer.